

# Benchmark study using a multi-scale, multi-methodological approach for the petrophysical characterization of reservoir sandstones

Peleg Haruzi[1,2], Regina Katsman[1], Matthias Halisch[3], Nicolas Waldmann[1], and Baruch Spiro[1,4]

[1] The Dr. Moses Strauss Department of Marine Geosciences, Faculty of Natural Sciences, The University of Haifa, Haifa, Mount Carmel 3498838, Israel

[2] Agrosphere Institute, IBG-3, Institute of Bio- and Geosciences, Forschungszentrum Jülich GmbH, Germany

[3] Leibniz Institute for Applied Geophysics, Dept. 5 – Petrophysics & Borehole Geophysics, Stilleweg 2, D-30655 Hannover, Germany

[4] Department of Earth Sciences, Natural History Museum, Cromwell Road, London SW7 5BD, UK

*Correspondence to:*     Regina Katsman (rkatsman@univ.haifa.ac.il)

Matthias Halisch (Matthias.Halisch@leibniz-liag.de)

**Keywords:** multi-methodological approach, permeability, petrography, petrophysics, 3D imaging, pore-scale modelling, upscaling, benchmark study



**Abstract**

This paper presents a detailed description and evaluation of a multi-methodological petrophysical approach for the comprehensive multiscale characterization of reservoir sandstones. The suggested methodology enables the identification of Darcy-scale permeability links to an extensive set of geometrical, textural and topological rock descriptors quantified at the pore scale. This approach is applied to the study of samples from three consecutive sandstone layers of Lower Cretaceous age in northern Israel. These layers differ in features observed at the outcrop, hand specimen, petrographic microscope and micro-CT scales. Specifically, laboratory porosity and permeability measurements of several centimetre-sized samples show low variability in the quartz arenite (top and bottom) layers but high variability in the quartz wacke (middle) layer. The magnitudes of this variability are also confirmed by representative volume sizes and by statistical anisotropy analyses conducted on micro-CT-imaged 3D pore geometries. Two scales of porosity variability are revealed by applying variogram analysis to the top layer: fluctuations at 150 µm are due to variability in the pore size, and those at 2 mm are due to the occurrence of high- and low-porosity bands occluded by iron oxide cementation. This millimetre-scale variability is found to control the laboratory-measured macroscopic rock permeability. Good agreement between the permeability upscaled from the pore-scale modelling and the estimates based on laboratory measurements is shown for the quartz arenite (top) layer. The proposed multi-methodological approach leads to an accurate petrophysical characterization of reservoir sandstones with broad ranges of textural, topological and mineralogical characteristics and is particularly applicable for describing anisotropy at various rock scales. The results of this study also contribute to the geological interpretation of the studied stratigraphic units.



## 1. Introduction

Permeability is an effective property of a reservoir rock that varies enormously over a wide range of rock length scales, attributed to a hierarchy of dominant sedimentary depositional features (Norris and Lewis, 1991; Nordahl and Ringrose, 2008; Ringrose and Bentley, 2015). Permeability should thus be properly upscaled through the following sequence of scales (Nordahl and Ringrose, 2008; Ringrose and Bentley, 2015 and references therein): (1) from the pore scale (the micro scale, typically microns to millimetres) to the representative elementary volume of a single lamina (the macro scale, typically millimetres to centimetres, e.g., Wildenschild and Sheppard, 2013; Andrä et al., 2013; Bogdanov et al., 2011; Narsilio et al., 2009); (2) to the scale of geological heterogeneity, e.g., the scale of a stratigraphic column (decimetres to decametres, e.g., Jackson et al. 2003; Nordahl et al. 2005); and (3) to the field scale or the scale of an entire reservoir or aquifer (hundreds of metres to kilometres) (Haldorsen and Lake 1984; Rustad et al., 2008). Pore-scale imaging and modelling enable us to relate macroscopic permeability to basic microscopic rock descriptors (Kalaydjan, 1990; Whitaker, 1986; Cerepi et al., 2002; Haoguang et al., 2014; Nelson, 2009). Therefore, the first stage in the above sequence is crucial for successful upscaling to the final reservoir-scale permeability.

Over the past few decades, pore-scale imaging and flow simulations (Bogdanov et al., 2012; Blunt et al., 2013; Cnudde et al., 2013; Wildenschild and Sheppard, 2013; Halisch, 2013) have started to serve as a reliable method for rock characterization. The advantages of these techniques are their non-destructive character and their capability to provide reliable information about the real pore-space structure and topology of rocks. However, despite its importance, the upscaling from the pore scale is sometimes omitted; as a result, effective petrophysical rock characteristics (e.g., porosity and permeability) are often evaluated at the macro scale through only conventional laboratory experiments, which often suffer from errors due to local heterogeneities or a small number of samples (e.g., Halisch, 2013).

The present paper provides a detailed description and evaluation of a multi-methodological petrophysical approach for the comprehensive multiscale characterization of reservoir sandstones. The proposed approach includes petrography, gas porosimetry and permeametry, mercury intrusion porosimetry, 3D imaging and several kinds of pore-scale modelling. The suggested computational workflow enables the identification of Darcy-scale permeability links to an extensive set of geometrical, textural and topological



rock descriptors, quantified at the pore scale by deterministic and probabilistic (statistical) methods.
Ultimately, this approach is applied to the study of three different consecutive sandstone layers of Lower
Cretaceous age in northern Israel.
The approach presented herein is especially important for the detection of anisotropy and the
identification of its origin at various rock scales. The multi-methodological validation procedure is
significant for properly upscaling permeability from the micro scale to the macro scale (Ringrose and
Bentley, 2015). This validation, thereby, allows an accurate petrophysical analysis of reservoir sandstones
with broad ranges of textural and topological characteristics. The findings contribute also to the current
geological knowledge regarding non-marine sandstones of Lower Cretaceous age (e.g., Akinlotan, 2017; Li
et al., 2016; Ferreira et al., 2016) and specifically regarding the studied stratigraphic unit.

**2.  Geological setting**
The study is based on samples collected from a steep outcrop at Wadi E'Shatr near Ein Kinya on the
southern slopes of Mt. Hermon (Fig. 1). The outcrop consists of sandstones from the Lower Cretaceous
Hatira Formation (Sneh and Weinberger, 2003). This formation (Fm.) acts as a reservoir rock for
hydrocarbons in Israel (Fig. 1a), both onshore, namely, Heletz (Grader and Reiss, 1958; Grader, 1959;
Shenhav, 1971), and offshore, namely, Yam Yafo (Gardosh and Tannenbaum, 2014; Cohen, 1971; Cohen,
1983; Calvo, 1992; Calvo et al., 2011).
The Hatira Fm. is the lower part of the Kurnub Group of Lower Cretaceous (Neocomian – Barremian)
age. The Kurnub Group in the study area (Fig. 1b, d) consists of a volcanic sequence at its base that is
overlain with an angular uncomformity by sandstone and clay layers of the Hatira Fm.; the upper unit
consists of limestone, marl and chalk – the Nabi Said Fm. (Sneh and Weinberger, 2004). At the section of
Saltzman (1967), which is approximately 100 m SW of the sampling area of the present study, the 58 m
thick variegated sandstone is interbedded with layers of clay and clay-marl. The sandy component is white-
yellow-brown/red and consists of largely angular, poorly sorted, fine- to coarse-grained quartz sand.
Individual sandstone layers are cemented by Fe-ox. The outcrops show lenticular benches 0.2 m -1.0 m
thick. The clay-rich interlayers are grey and normally siltic and brittle. Locally, these layers contain lignite.





The wider geological context of the Hatira Fm. is presented in Appendix A. The outcrop investigated and
the specific beds sampled in the present study are shown in Figure 1c.

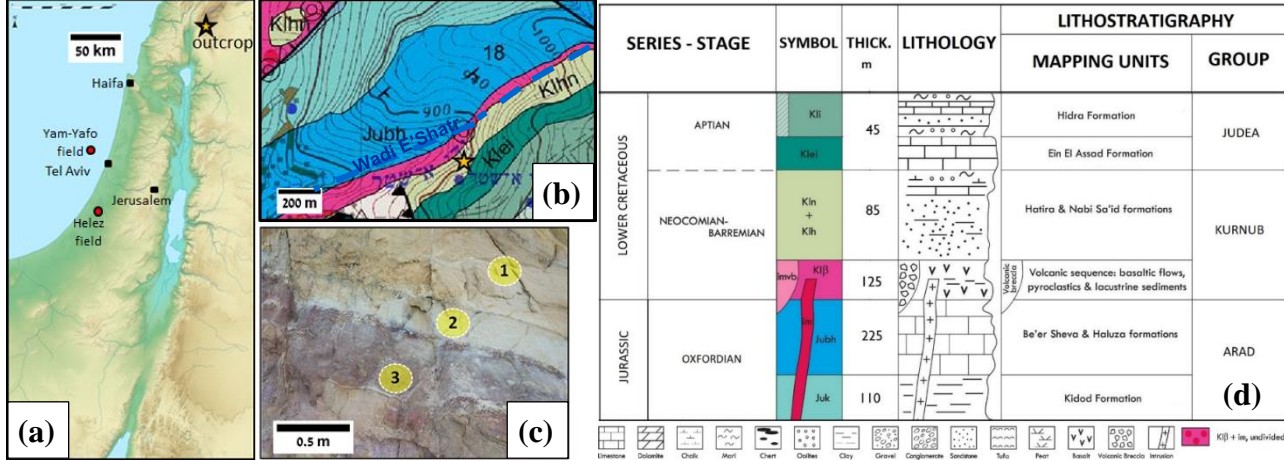


***Figure 1:*** *Geographical and geological settings. (a) Schematic relief map of Israel. The site of Ein Kinya on*
*the southern flanks of Mt. Hermon is indicated by a star (modified from www.mapsland.com). (b) Geological*
*map of Ein Kinya. The Hatira Fm. sandstone and the overlying limestone and marl of the Nabi Said Fm. are*
*marked as Klhn (map is adopted from Sneh and Weinberger, 2014). (c) Outcrop of the* Lower Cretaceous
*Hatira Fm. sandstones (Klhn) at Ein Kinya. The studied sandstone layers have distinct colours: yellow-*
*brown (1), grey-green (2), and red-purple (3). (d) Stratigraphic table of the geological map (modified from*
*Sneh and Weinberger, 2014).*

**3.   Methods**
**3.1. Sample description**

Samples were extracted from three consecutive layers of different colours from a stratigraphic

sequence (Figs. 1c, 1d). The lower layer (3) is ~1.5 m thick and consists of sandstone that is light (pale) red-
purple in colour with undulating bedding planes between the sub-layers. The middle layer (2) is composed of
grey – green shaly sandstone that is 20 cm thick with dark horizons at the bottom and top. The upper layer





(1) comprises 1.5 m thick homogenous brown-yellow sandstone. Large sample blocks were collected from
these three layers, and the directions perpendicular to the bedding planes (defined as the z-directions in our
study) were noted. Subsequently, in the laboratory, smaller sub-samples (described below) were prepared
from these large samples for textural observations and various analytical measurements and computations. In
total, 7 sub-samples from the top layer, 8 sub-samples from the middle layer and 4 sub-samples from the
bottom layer were investigated in the laboratory (Table 2).
**3.2. Laboratory and computational methods for rock characterization**
The integrated analytical programme designed for this study includes the following laboratory
measurements and computations conducted at different scales (from the micro scale reflecting the scale of
individual pores and grains to the core scale reflecting the scale of the laminas at the outcrop) (Table 1).
Specimens ~5-7 cm in size were investigated by petrographic and petrophysical lab methods. Sub-samples
~1 cm in size were retrieved from the aforementioned plugs for investigation by 3D imaging, digital image
analysis and simulation techniques (described in more detail below).
*Table 1.* *Laboratory methods employed and petrophysical characteristics determined in this study*

| Method | Determined petrophysical characteristics |
|---|---|
| 1. Scanning electron microscopy (SEM) | Mineral abundance, grain surface characterization of matrix and cementation |
| 2. Grain size analysis (Laser diffraction) | Grain size distribution (*GSD*) |
| 3. X-ray diffraction (XRD) | Mineral components |
| 4. Nitrogen gas porosimetry | Porosity ($\phi$) |
| 5. Steady state permeametry | Permeability (1D) ($\kappa$) |
| 6. Mercury intrusion porosimetry (MIP) | Pore throat size distribution (*PTSD*), specific surface area (*SSA*), characteristic length ($l_c$), pore throat length of maximal conductance ( $l_{max}$ ), permeability ($\kappa$) |
| 7. Petrographic microscopy Plane-parallelized (PPL) and cross-parallelized (XPL) and reflected-light (RL) microscopy, binocular (BINO). | Mineral abundance, grain surface characterization, cementation |





| 8. Extended computational workflow: | |
|---|---|
| Digital image analysis (DIA) | Porosity ($\phi$), pore specific surface area (*PSA*), tortuosity ($\tau$), pore size distribution (*PSD*), connectivity index (*CI*), micro-CT predicted porosity from MIP |
| Fluid flow modelling | Permeability tensor ($\bar{\bar{\kappa}}$), tortuosity ($\tau$) |

Petrographic descriptions of rock compositions and textures at the micro scale, notably those of the fine fraction, were performed using scanning electron microscopy (*JCM-6000 Bench Top SEM device*, Krinsley et al., 2005) using both backscatter and secondary electron modes.

Thin-section optical microscopy (*Olympus BX53 device*, Adams et al., 2017) was used to estimate the mineral abundance and surface features of the grains, and the mineralogical and textural features of matrix and cement. Grain size distributions were determined by a laser diffraction particle size analyser (LS 13 320). X-ray diffraction (*Miniflex 600 device by Rigaku*) was applied to powdered samples to determine their mineralogical composition.

Effective porosity and permeability were evaluated on dried cylindrical samples (2.5 cm in diameter and 5-7 cm in length) following the RP40 guidelines (*Practices for Core Analysis, API, 1998*). Effective porosity ($\phi$) was measured using a steady-state nitrogen gas porosimeter produced by Vinci Technologies (*HEP-E, Vinchi Technology,* v3.20). Absolute permeability ($\kappa$) was measured by using a steady-state nitrogen gas permeameter (*GPE, Vinci Technologies*; e.g., Tidwell and Wilson, 1999).

Mercury intrusion porosimetry (*Micromeritics AutoPore IV 9505*, which considers pore throats larger than 0.006 μm) was applied to dried cylindrical samples ~1 cm$^3$ in size to evaluate the following parameters (Table 1):

- Pore throat size distribution (*PTSD,* Lenormand, 2003).
- Specific surface area (*SSA*): the pore surface to bulk sample volume (Rootare and Prenzlow, 1967; Giesche, 2006).
- Characteristic length ($l_c$): the largest pore throat width (obtained from the increasing intrusion pressure) at which mercury forms a connected cluster (Katz and Thompson, 1987).





•   Pore throat length of maximal conductance ($l_{max}$): defines a threshold for the pore throat size l at

which all connected paths composed of $l \geq l_{max}$ contribute significantly to the hydraulic

conductance, whereas those with $l < l_{max}$ may completely be ignored (Katz and Thompson,

1987).

•   Permeability (Katz and Thompson, 1987):
$$\kappa = \frac{1}{89} l_{max}^2 \frac{l_{max}}{l_c} \phi S(l_{max}) \tag{1}$$

where $S(l_{max})$ is the fraction of connected pore space that is composed of pore throat widths of size $l_{max}$ and
larger. This approach (Katz and Thompson, 1987), which was derived from percolation theory (Ambegaokar
et al., 1971), is applicable for sandstones with a broad distribution of local conductances with short-range
correlations only.
An extended computational workflow (similar to the procedure presented by Boek and Venturoli,
2010; Andrä et al., 2013) (Fig. 2) serves as one of the main methodologies in our study to upscale
permeability. It includes 3D micro-CT imaging of porous samples, digital image processing and
segmentation, statistical analyses for the determination of representative elementary volumes, and pore-scale
flow modelling through the 3D pore geometry of the rock. First, cylindrical subsamples 4-8 mm in diameter
and 5-10 mm in length were retrieved from the larger samples studied in the laboratory and were scanned
non-destructively (Fig. 2b) by using a *Nanotom* 180 S micro-CT device (*GE Sensing & Inspection*
*Technologies, phoenix|X-ray product line*, Brunke et al., 2008). The achieved voxel size of the data sets was
2.5 µm (isotropic), suitable for imaging pore throats that effectively contribute to the flow in the studied type
of sandstone (e.g., Nelson, 2009). Afterwards, all data sets were filtered for de-noising, X-ray artefact
removal and edge enhancement (Fig. 2c). The post-processed images had an edge length of 1180 voxels or
2950 µm. Image artefacts were processed as described by Wildenschild and Sheppard (2013). Beam
hardening artefacts were removed by applying the best-fit quadratic surface algorithm (Khan et al., 2016) to
each reconstructed 2D slice of the image. Ring artefact reduction and image smoothing (with preservation of
sharp edge contrasts) were performed using a non-local means filter (Schlüter, 2014). Segmentation was
performed to convert the grey-scale images obtained after image filtering into binary images to distinguish
between voids and solid phases (Fig. 2c). The local segmentation approach, which considers the spatial
dependence of the intensity for the determination of a voxel phase, was used in addition to a histogram-based





approach (Iassonov et al., 2009; Schlüter et al., 2014). Two-phase segmentation was performed by the
converging active contours algorithm (Sheppard et al., 2004), a combination of a watershed (Vincent et al.,
1991) with an active contour algorithm (Kass et al., 1988).


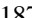


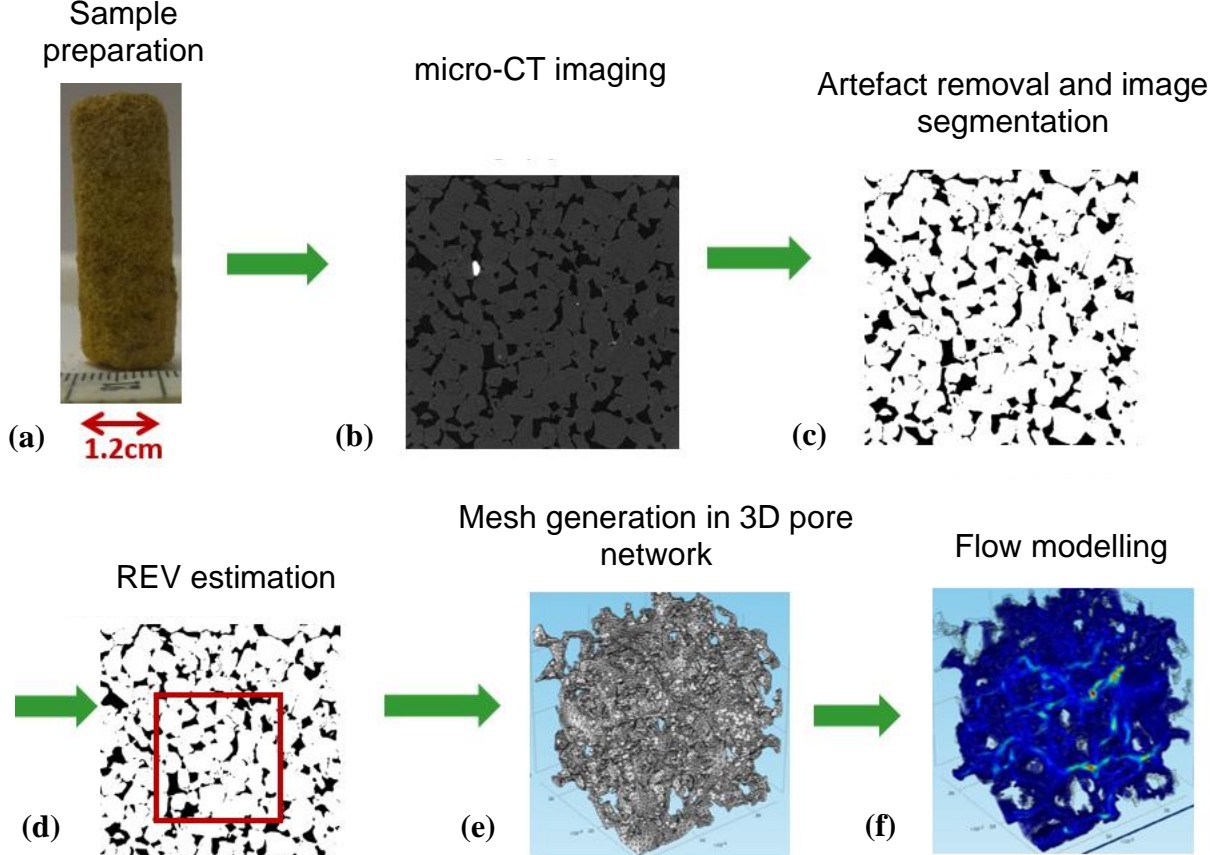

**Figure 2.** *Extended computational workflow. See text for more details. Images (**e**) and (**f**) are adopted from Bogdanov et al. (2012).*

Simulations involving the real geometry of an imaged rock are computationally power and time consuming. Therefore, the determination of a representative elementary volume (REV) is required (Fig. 2d), assuming that porous media are homogeneous at REV dimensions (Bear, 1988). An REV is required in the current study to perform fluid flow simulations. Porosity, a basic macroscopic structural property of porous media, is used here for the estimation of an REV (Bear, 1988; Halisch, 2013; Tatomir et al., 2016) based on its correlation with permeability (Kozeny, 1927; Carman, 1956).

Two approaches were used in this study to estimate the REV (Halisch, 2013). In the "classic" approach, the REV is attained when porosity fluctuations in the sub-volumes that grow isotropically in three



orthogonal directions become sufficiently small (Bear, 1988). Practically, a large number of randomly
distributed cubes were analysed through the entire 3D sample (with a 1180 voxel edge length in our case) for
their image porosity (IP). The chosen initial cube size (with an edge length of 10 pixels in our case) was
increased by 10-100 voxels. The REV size was specified when agreements between the mean and median IP
values as well as saturation in the IP fluctuations were attained. The results of the REV estimation by this
classic approach can be found in Appendix B.

A more advanced "directional" REV approach can capture porosity changes in a specific direction

caused by microscopic structural features, such as grain packing, cracks, and textural effects (Halisch, 2013).
The average porosity is first calculated slice by slice across the segmented image in each orthogonal
direction. Variogram analysis (Cressie, 1993) is used to describe the degree of spatial variability of the
porosity in each direction based on the assumption that a distance at which no spatial correlation exists
reflects the scale of homogeneity, which defines the REV. The variogram $\hat{\gamma}(h)$, i.e., the expected squared
difference between two observations (here, the average of 2D IPs), is calculated as a function of their
separation distance, *h (lag)*. Practically, the lag distance at which the variogram curve is saturated is the
distance at which no spatial correlation exists (defined as the range of a spatial correlation). Depending on
the sample heterogeneity at different scales, the variogram may manifest a different range for each scale.
Variogram analysis was performed using the 'Variogramfit' MATLAB package.

Further, the representative binary 3D image (REV) of the pore space was spatially discretised by

tetrahedrals with *Materialize software (Belgium)* (Fig. 2e). This step is required for importing the REV into
the FEM-based modelling software (*Comsol Multiphysics simulation environment*, v5.2a). Stokes flow (Re
<< 1) is simulated (Table 1) in the pore network (Fig. 2f) by the following equations (e.g., Narsilio et al.,
2009; Bogdanov et al., 2011):

Stokes equation:        $-\nabla p + \mu \nabla^2 \bar{u} = 0$                    (2)

Continuity equation:     $\nabla \cdot \bar{u} = 0$                              (3)

where $\nabla p$ is the local pressure gradient, $\bar{u}$ is the local velocity vector in the pore space and μ is the dynamic
fluid viscosity. Fixed pressures (*p=const*) were specified at the inlet and outlet boundaries of the fluid
domain. At the internal pore walls and at the lateral domain boundaries, no-slip boundary conditions ($\bar{u} = 0$)
were imposed (e.g., Guibert et al., 2014). These also simulate the flow setup in a steady-state experimental



permeameter (e.g., Renard et al., 2001). The macroscopic fluid velocity $< \bar{v} >$ was evaluated by
volumetrically averaging the local microscopic velocity field (e.g., Narsilio, 2009; Guibert et al., 2016).
Then, from the average macroscopic velocity vectors $v_i^j$ in three orthogonal *i*-directions corresponding to the
pressure gradients $\nabla p_j$ imposed in *j*-directions, the full 3D second-rank upscaled permeability tensor $\bar{\bar{\kappa}}$ can
be found:
$$\begin{pmatrix} v_x^x & v_x^y & v_x^z \\ v_y^x & v_y^y & v_y^z \\ v_z^x & v_z^y & v_z^z \end{pmatrix} = -\frac{1}{\mu\phi} \begin{pmatrix} \kappa_{xx} & \kappa_{xy} & \kappa_{xz} \\ \kappa_{yx} & \kappa_{yy} & \kappa_{yz} \\ \kappa_{zx} & \kappa_{zy} & \kappa_{zz} \end{pmatrix} \begin{pmatrix} \nabla p_x & 0 & 0 \\ 0 & \nabla p_y & 0 \\ 0 & 0 & \nabla p_z \end{pmatrix}$$
(4)

The permeability tensor is symmetrized by:
$$\bar{\bar{\kappa}}_{sym} = \frac{1}{2}(\bar{\bar{\kappa}} + \bar{\bar{\kappa}}^T)$$
(5)

Tortuosity ($\tau$; Bear, 1988; Boudreau, 1996) was calculated separately in the x-, y- and z-directions in
the meshed domain using the particle tracing tool of *Comsol Multiphysics software* (an additional method for
deriving $\tau$ is presented later in this section).
3D image analysis (Table 1) was conducted on a high-quality, fully segmented micro-CT image (edge
length of 2950 µm scanned at a 2.5 µm voxel size). Non-connected void clusters in the binary specimen were
labelled and then separated into objects (single pores and grains) by using a distance map followed by the
application of a watershed algorithm (e.g., Brabant et al., 2011; Dullien, 2012). Image analysis operations
were assisted by *Fiji-ImageJ software* (Schindelin et al., 2012) and by the *MorphoLibJ plug-in* (Legland et
al., 2014). The following geometrical descriptors were derived from the segmented image limited by the
image resolution of 2.5 µm (Table 1):
• micro-CT image porosity (*IP*);
• Pore specific surface area (*PSA* – surface to pore volume);
• Tortuosity: evaluated in the x-, y- and z-directions by finding the average of multiple shortest paths
through the main pore network using the fast marching method (Sethian, 1996) implemented using
an accurate fast marching plug-in in MATLAB.
• Pore size distribution *(PSD)*: obtained by a Feret maximum calliper (Schmitt et al., 2016).





- Euler characteristic ($\chi$) - a topological invariant (Wildenschild and Sheppard, 2013; Vogel, 2002) that describes the structure of a topological space (see Appendix C for more detail). Since the number of pore connections depends on the number of grains, it is essential to normalize $\chi$ (Scholz et al., 2012) to compare the connectivity among three samples that have the same dimensions but different grain sizes.

- Connectivity index (*CI*): computed by dividing the absolute value of the Euler characteristic ($|\chi|$) by the number of grains in the specimen (*N*, determined by image analysis), $CI = |\chi|/N$.

Additionally, we propose a simple and new method to estimate the image porosity at a given resolution. Multiplication of the mercury effective saturation at the capillary pressure corresponding to the micro-CT resolution (i.e., 2.5 µm) by the porosity of the same sample measured by a gas porosimeter yields the *micro-CT-predicted image porosity from MIP* at the given resolution limit (Table 1).





## 4. Results

### 4.1. Petrographic and petrophysical rock characteristics

Three types of sandstone rocks were characterized by techniques 1-8 listed in Table 1. The results are presented in Figures 3-8 and summarized in Table 2.

**Sandstone S1**: The top unit layer with a thickness of ~1.5 m (Fig. 1c) consists of yellow-brown sandstone (Fig. 3a), which is moderately consolidated. The sandstone is a mature quartz arenite (following Pettijohn et al., 1987) with minor Fe-ox, feldspar and heavy minerals. The grain size distribution has a mean of ~325 µm (Fig. 6a, Table 2). The grains are moderately sorted (according to the classification of Folk and Ward, 1957) and sub-rounded to well-rounded with local thick (millimetre-scale), relatively dark envelopes (Fig. 3b). The sandstone consists of alternating millimetre-scale layers of large and small sand grains. Secondary silt (~ 45 µm) and clay (~0.95 µm) populations are detected in the grain size distribution (Fig. 6). X-ray diffraction detected a small amount of kaolinite. The Fe-ox grain-coating and meniscus-bridging cement is composed of overgrown flakes aggregated into structures ~10 µm in size (Fig. 3c-3f). Mn-ox is also evident but is scarce (Fig. 3e).

The pore network is dominated by primary inter-granular well-interconnected macro porosity (Fig. 3b). However, sealed and unsealed cracks in grains are also observed. Higher Fe-ox cementation at the millimetre scale on horizontal planes is recognized (Fig. 3a). In addition, smaller voids between Fe-ox aggregates and flakes occur at the micrometre scale and smaller (Fig. 3d-f).

The pore throat size analysis conducted with MIP shows that 82 % of the pore volume is composed of macro pores (>10 µm) following a log-normal distribution with a peak at 44 µm (Fig. 7a). The characteristic length, i.e., the largest pore throat length at which mercury forms a connected cluster, is $l_c = 42.9$ µm (Fig. 7b), and the pore throat length of maximal conductance is $l_{max} = 34.7$ µm (Appendix D, Fig. D1). The porosity evaluated by laboratory gas porosimetry varies in the range of 26-29 % for 7 different samples of S1 (Fig. 8). Multiplying the mercury effective saturation (85.8 %) at the micro-CT resolution (2.5 µm) (Fig. 7a, red dashed line) by the porosity of the same sample measured by gas porosimetry (27.3 %) yields a micro-CT-predicted image porosity of 23.5 % at a resolution limit of 2.5 µm (Table 2).





The permeability evaluated by a laboratory gas permeameter has averages of 350 mD (range of 130-
500 mD) for 5 samples measured perpendicular to the depositional plane (z-direction) and 640 mD for 2
samples measured parallel to the depositional plane (x- and y-directions) (Fig. 8). MIP measurement (Katz
and Thompson, 1987) yields a permeability (see Sect. 3.2) of 330 mD (Table 2).

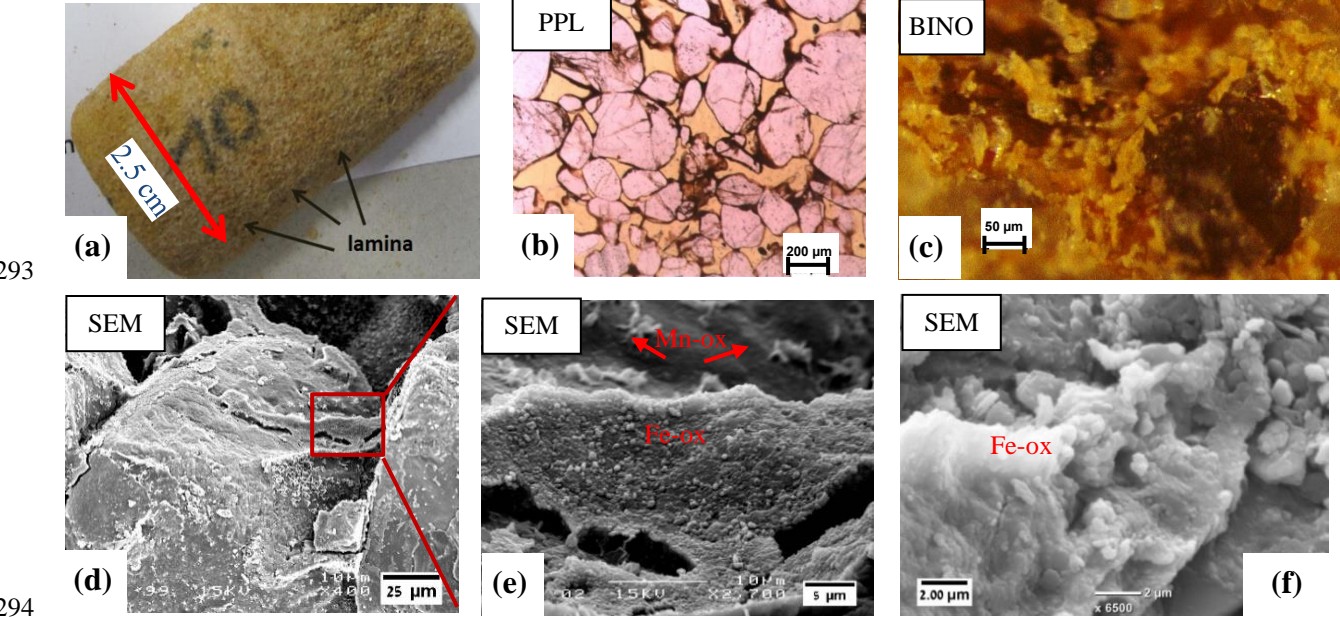




*Figure 3: Representative images of sandstone S1. (a) Darker laminae in the x-y plane at the millimetre scale*
*are observed. (b) Thin section image of S1: quartz grains are shown in pink, while pores are in yellow. (c)*
*Fe-ox flakes (yellow) on quartz grains (pale grey) identified with binocular. (d) SEM image of S1: grain-*
*coating, meniscus-bridging cement and overgrowth of Fe-ox flakes are observed. Magnified images at*
*different scales are presented in (e)-(f).*





**Table 2.** *Petrophysical characteristics of the three studied sandstone layers.*

| | Method | S1 | S2 | S3 |
|---|---|---|---|---|
| **Grain size** | Laser diffraction | 325 µm<br>**medium sand**<br>**moderately sorted**<br>sand: 92.6 %<br>silt: 6.6 %<br>clay: 0.8 % | 154 µm<br>**very fine sand**<br>**poorly sorted**<br>65.7 %<br>31.3 %<br>3 % | 269 µm<br>**fine sand**<br>**moderately sorted**<br>94.4 %<br>4.8 %<br>0.8 % |
| **Pore throat size** | MIP | Mode 1: 44 µm<br>Mode 2: 0.035 µm<br>Mode 3: 2.2 µm<br>**macro pores**<br>**well sorted** | 0.035 µm<br>3.5 µm<br>**meso pores**<br>**poorly sorted** | 35 µm<br>0.035 µm<br>2.2 µm<br>**macro pores**<br>**well sorted** |
| **Pore size** | Image analysis (min. object size 2.5 µm) | 194 µm<br>(*FWHM [150,335] µm) | Mode 1: 21 µm<br>Mode 2: ~100 µm | 223 µm<br>(*FWHM [145,400] µm) |
| **Characteristic length, $l_c$** | MIP | 42.9 µm | 12.3 µm | 36.9 µm |
| **$l_{max}$ contributing to maximal conductance** | MIP | 34.7 µm | 8 µm | 31.4 µm |
| **Porosity, ϕ** | Gas porosimetry | 28 ± 2 % (7**) | 19 ± 5 % (8) | 31 ± 1 % (4) |
| | CT predicted image porosity from MIP | 23.5 % | 6.6 % | 30.4 % |
| | Micro-CT segmented | 17.5 % | 6.9 % | 28.3 % |
| **Permeability, $\kappa$**<br>⊥ - perpendicular to layering (z-direction)<br>\|\| - parallel to layering (x-y plane) | Gas permeametry | ⊥ 350 mD (5)<br>\|\| 640 mD (2) | ⊥ 2.77 mD (5)<br>\|\| 7.73 mD (3) | ⊥ 220 mD (2)<br>\|\| 4600* mD (2) |
| | MIP | 330 mD (1) | 4 mD (1) | 466 mD (3) |
| | Flow modelling | $\begin{pmatrix} 420 & 66.3 & 1.91 \\ 66.3 & 344 & 12.8 \\ 1.91 & 12.8 & 163 \end{pmatrix}$ mD | - | $\begin{pmatrix} 4517 & 5 & 38 \\ 5 & 4808 & 547 \\ 38 & 547 & 4085 \end{pmatrix}$ mD |
| **Specific surface area, SSA** (surface-to-bulk-volume) | MIP | 3.2 $\mu m^{-1}$ | 12.2 $\mu m^{-1}$ | 0.16 $\mu m^{-1}$ |
| **Pore specific surface area, PSA** (surface-to-pore-volume) | Micro-CT at 2.5 µm resolution size | 0.068 $\mu m^{-1}$ | 0.136 $\mu m^{-1}$ | 0.069 $\mu m^{-1}$ |
| **Connectivity index** | Image analysis | 3.49 | 0.94 | 10 |
| **Tortuosity, $\tau$** | Flow modelling | - | - | x: 1.443<br>y: 1.393<br>z: 1.468 |
| | Micro-CT shortest path analysis | x: 1.385<br>y: 1.373<br>z: 1.477 | - | x: 1.316<br>y: 1.338<br>z: 1.394 |

Legend:





*Addressed in the Discussion.
** Numbers in parentheses related to gas porosity, gas permeability and MIP permeability indicate the
number of plugs for the measurements. Other measurements and calculations were conducted on single
plugs.
FWHM - full width at half maximum, log-normal distribution.

**Sandstone S2**: The intermediate unit layer with a thickness of ~20 cm consists of grey-green
moderately consolidated sandstone (Figs. 1c, 4) composed of sub-rounded to rounded, very fine sand grains
(~154 μm); the sandstone is poorly sorted with 35 % of the particles being silt and clay (Fig. 6, Table 2).
Secondary silt (~ 40 μm), sand (~400 μm) and clay (~1.5 μm) populations are also detected. The grains are
composed of quartz with minor Fe-ox coating the grains and minor quantities of heavy minerals (Fig. 4c).
Clay filling the pore space was identified by XRD as a kaolinite mineral. It appears as a grain-coating,
meniscus-bridging, and pore-filling matrix (Fig. 4b, c). Therefore, the unit layer (Fig. 1c) is classified as a
quartz wacke sandstone.
The pore network is influenced by the extent of clay deposition on coarser grains, identified mostly in
laminae (Fig. 4a, d). However, the inter-granular connectivity of macro pores can still be recognized (Fig.
4b, c). The effective pore network consists of inter-granular macro pores distributed between the laminae or
zones richer in clay and Fe-ox. Integrating the grain size and pore throat size analysis results (Figs. 6, 7)
confirms that the reduction in the inter-granular pore space in S2 is due to the clay matrix, which is reflected
in the poor grain sorting and large variance in pore size. In the pore throat size analysis (Fig. 7), only 15 %
of the pore volume is composed of macro pores that are larger than 10 μm. The prominent sub-micron pore
mode is ~35 nm, with a population containing ~45 % of the pore volume (Fig. 7a). This population of pores
occurs inside the clay matrix. The secondary pore volume population is poorly distributed within the range
of 0.8-30 μm. The characteristic length (Sect. 3.2), $l_c = 12.3$ μm (Fig. 7b), and the pore throat length of
maximal conductance, $l_{max} = 8$ μm (Appendix D, Fig. D1) (both have a large uncertainty resulting from
uncertainty in the threshold pressure), suggest a connectivity of macro pores regardless of their small
fraction within the total pore space. The porosity of S2 evaluated for 8 different samples varies in the range
of 14.5-23.5 % (Fig. 8). From the *PTSD* (Table 1) and gas porosimetry results (for a sample with a porosity
of 18.6 %), micro-CT predicts an image porosity of 6.6 % at a resolution limit of 2.5 μm (Table 2). The gas





permeability in the z-direction was measured in 5 samples (Fig. 8): in four of them, the permeability ranges within 1-12 mD and increases with porosity. However, one sample had an exceptionally large porosity and permeability of 23 % and 62 mD, respectively. The permeability measured for 3 samples in the x-y plane ranges within 4-12 mD, also showing ~15 % porosity (Fig. 8). In addition, for the samples with ~15 % porosity, their permeability is ten times larger in the x-y plane (parallel to the layering) than in the z-direction (perpendicular to the layering). The permeability derived from MIP reaches 4 mD, which agrees with an average of 2.77 mD and 7.73 mD (Table 2) measured in the z-direction with a gas permeameter (excluding one exceptionally high value, Fig. 8).

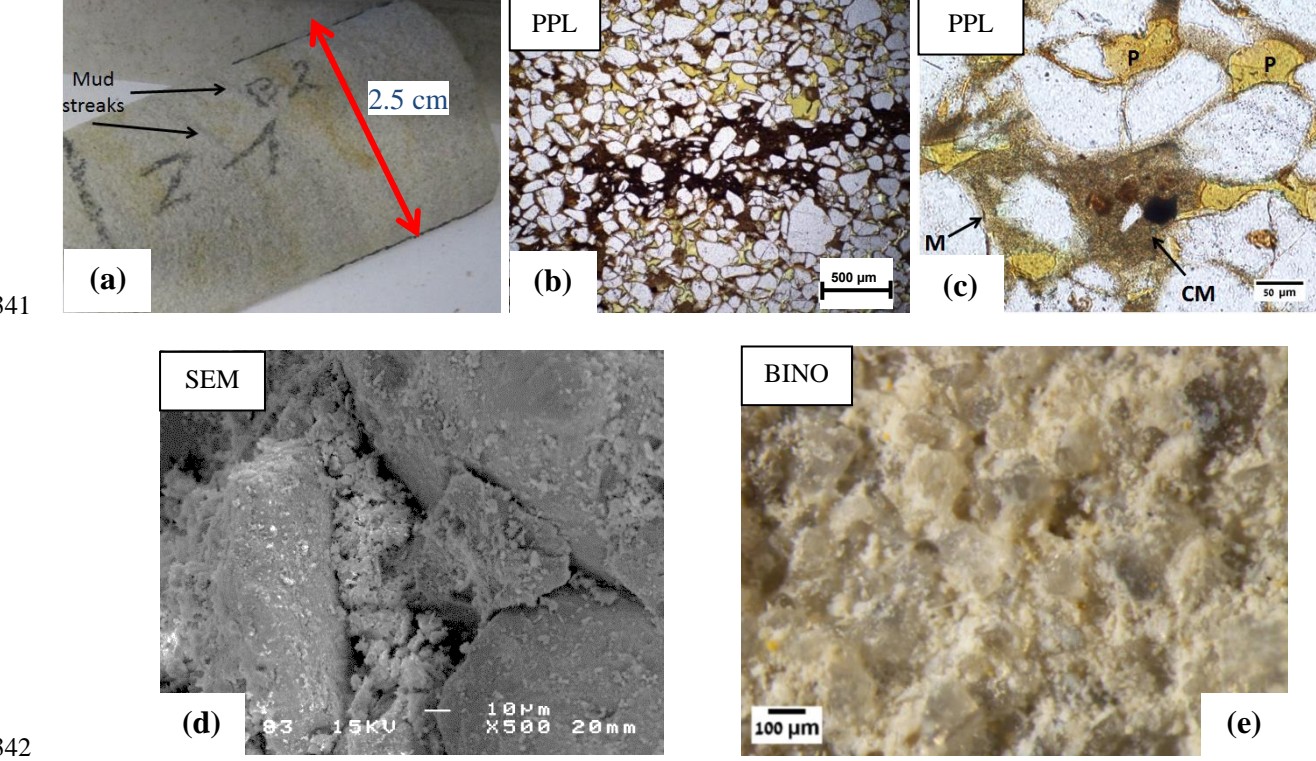

**Figure 4.** *Representative images of sandstone S2. (a) Prominent dark and yellowish zones are observed. (b) The dark laminae are richer in clays and Fe-ox. (c) Clay and silt accumulated as meniscus (M) and as clay matrix (CM). P refers to open pores. (d) Pore clogged by clay and Fe-ox. (e) Rock texture under binocular. The clay matrix is white, and quartz grains are pale grey.*





**Sandstone S3**: Samples were taken from the ~1.5 m thick bottom unit layer in the outcrop (Fig. 1c)
consisting of (pale) red-purple poorly consolidated sandstone with grains covered by a secondary red patina
(Fig. 5). The sandstone is composed of friable to semi-consolidated, fine (~269 µm), moderately sorted sand
(Table 2), where only 5.6 % of particles are silt and clay (Fig. 6). Secondary silt (~ 50 µm) and clay (~ 0.96
µm) populations were also detected. The sandstone consists of sub-rounded to rounded grains showing a
laminated sedimentary texture consisting of the cyclic alternation of relatively dark and light red bands of
millimetre-scale thickness (Fig. 5a). The dark laminae contain slightly more Fe-ox meniscus-bridging and
pore-filling cementation (Fig. 5b, d). Overall, this bed consists of a ferruginous quartz arenite. The grains are
dominated by quartz with very minor feldspar and black opaque mineral grains, perhaps Fe-ox (Fig. 5d). X-
ray diffraction indicated quartz only. The Fe-ox coating of grains is less extensive than in other samples
(Fig. 5c). The pore interconnectivity in this sandstone is high (Fig. 5d). Heavier cementation is rarely
observed (Fig. 5d) and is organized in horizontal laminae (Fig. 5a). Features including grain cracks, grain-
to-grain interpenetration, and pressure solution are also recognized (Fig. 5e). The *PTSD* showed that 95 % of
the pore volume is presented by macro pores (Fig. 7a), which agrees with the minority of fine particles. The
characteristic length and pore throat length of maximal conductance are $l_c$ = 36.9 µm (Fig. 7b) and $l_{max}$ =
31.4 µm (Appendix D, Fig. D1), respectively.
The porosity measured by a gas porosimeter in the laboratory varies in the range of 30-32 % for 4
different samples (Fig. 8). From *PTSD* and gas porosimetry (Figs. 7, 8), the micro-CT-predicted image
porosity at a resolution limit of 2.5 µm is 30.4 % (Table 2). The permeability measured by a laboratory gas
permeameter averages 220 mD for 2 samples measured in the z-direction and 4600 mD for 2 samples
measured in the x-y plane (Fig. 8), showing a ten-fold difference (discussed in Sect. 5). The permeability
derived from MIP reaches 466 mD (Table 2).


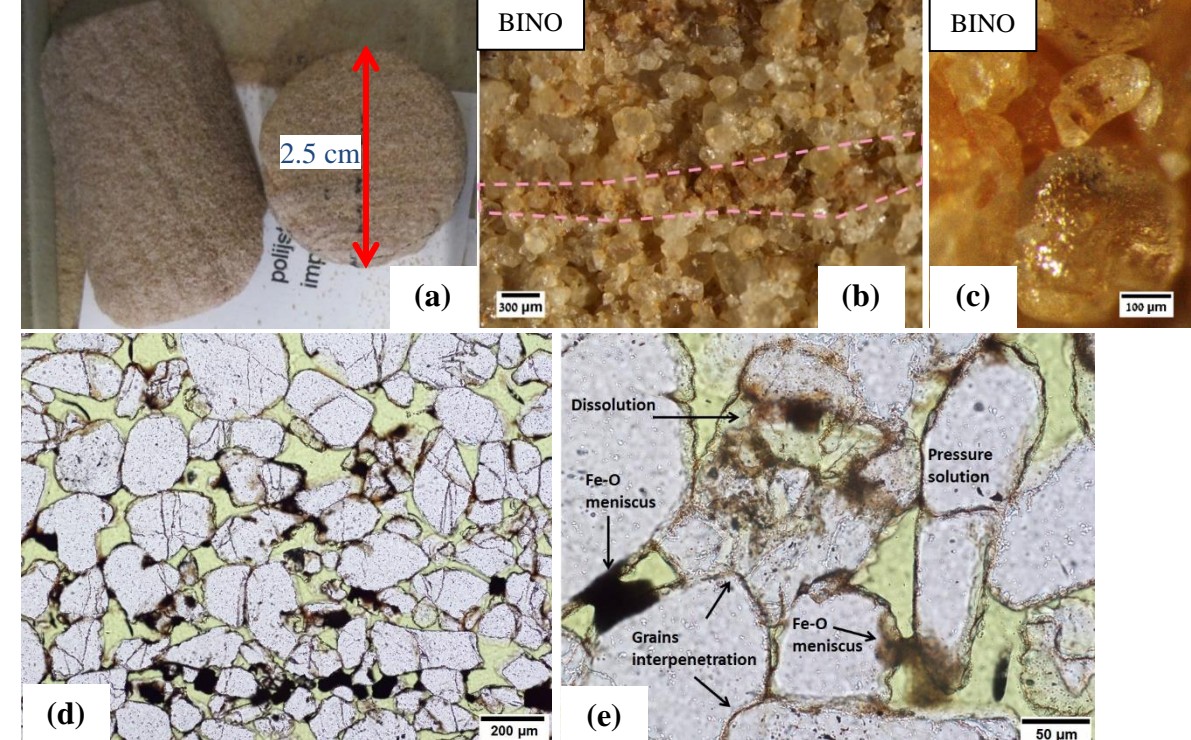

**Figure 5.** *Representative images of sandstone S3. (**a**) Laminae are recognized by their slightly dark and red colour. (**b**) General view under a binocular microscope reveals red laminae ~500 μm thick. (**c**) High-resolution observation of a clear grain under binocular. (**d**) A millimetre-scale lamina is indicated by enhanced meniscus-type Fe-ox cementation and partly by inter-granular fill. Grain surfaces are coated by thin Fe-ox. Black and orange cements represent crystallized and non-crystallized Fe-ox, respectively. Some cracked grains are observed, sporadically cemented by Fe-ox. (**e**) Partially dissolved grains are coated by cement.*


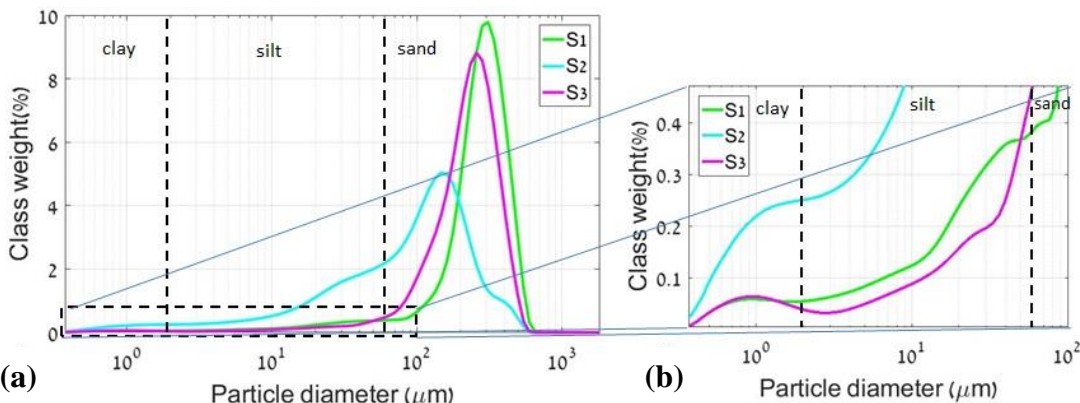


**Figure 6:** *(a) Grain size distribution. (b) Magnified grain size distribution in the fine grain size region plotted for sandstones S1 (green), S2 (blue) and S3 (purple). S1 and S3 have a unimodal distribution and are moderately sorted with a small skewness tail. Sample S2 has a multi-modal distribution and is poorly sorted.*

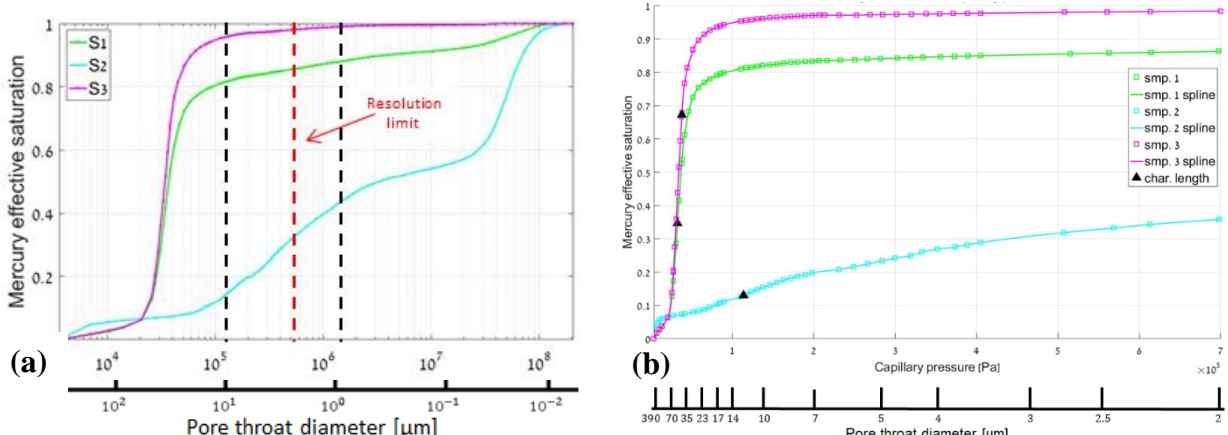


**Figure 7:** *Cumulative pore throat sizes of the studied sandstones. (a) Capillary pressure on a logarithmic scale. The resolution limit of the micro-CT imaging indicates the fraction of the pore space that could be resolved. (b) Capillary pressure on a normal scale. The triangles indicate the characteristic length, $l_c$.*






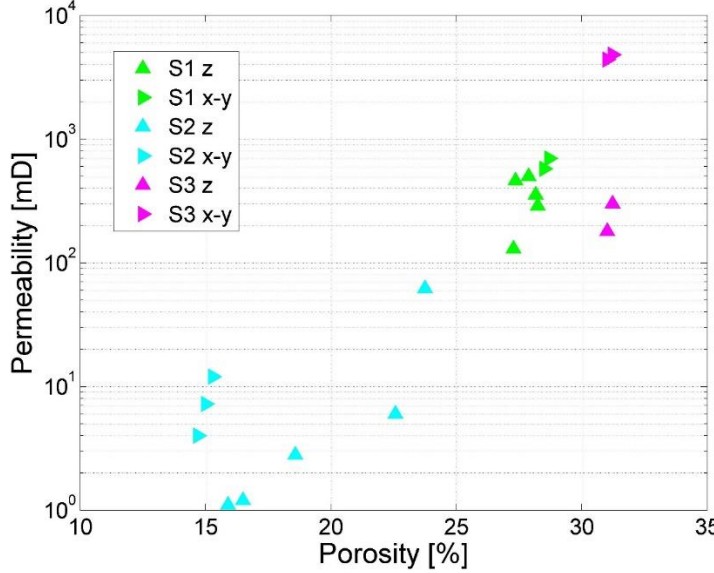


***Figure 8:*** *Results of porosity-permeability lab measurements. The permeability of the samples was measured in directions perpendicular to the bedding (z-direction) and parallel to the bedding (x-y plane).*

Overall, for all three investigated sandstones, the pore throat size contributing to the maximal conductance, $l_{max}$, is smaller than the characteristic length, $l_c$ (Table 2), when the relative decrease is greater for the layers containing more fines.

Additionally, pore surface roughness may be evaluated from the specific surface area (*SSA*) measured by MIP (Table 2). A larger *SSA* implies a rougher surface (e.g., Tatomir et al., 2016). The *SSA*s for S1 and S2 (3.2 μm$^{-1}$ and 12.2 μm$^{-1}$, respectively) are similar to those given in the literature for sandstones of similar properties (e.g., Cerepi et al., 2002). The *SSA* of S2 is higher because of its high silt and clay content of 34.3 %, which is only 7.4 % for S1 (Fig. 6a). The *SSA* of S3 (where silt and clay constitute only 5.6 %, including the Fe-ox rim coating) is only 0.16 μm$^{-1}$, which is 20 times smaller than that of S1 (Table 2). The difference in *SSA*s between S1 and S3, which are similar in their grain and pore throat size distributions (Figs. 6, 7), is a result of S1 having a higher Fe-Ox grain coating than S3 (compare Figs. 3d and 5c).





In summary, although the S1 pore network has larger pore throats, it also has greater grain roughness
and lower connectivity than S3. These two properties dominate and generate a smaller permeability for S1
than for S3 (Table 2).

**4.2. REV Analysis**
The results of the directional REV analysis of sample S1 conducted on a cube with a 2950 μm edge
size (1180 pixels) scanned with a resolution of 2.5 μm are shown in Figures 9 and 10. The average slice-by-
slice porosity analysed in three directions distinguishes the z-direction as having an exceptional behaviour
(Fig. 9a-c). Specifically, the difference between the maximum and minimum porosities is 7.44 % in the z-
direction (in contrast to 5.7 % in the x- and y-directions) with a standard deviation of 1.83 % (in contrast to
1.24 % and 1.01 % in the x- and y-directions, respectively). Along the z-direction (Fig. 9c), the porosity in
the domain below slice #250 is ~15 %, and that in the domain above slice #250 is ~18 %. There are cyclic
porosity fluctuations with a period of ~150 μm (60 pixels) associated with the size of a grain in each
domain. The variogram in the z-direction (Fig. 9f) shows a larger variability than those in the other two
directions, which refers to zonal anisotropy caused by layering (Gringarten and Deutsch, 2001). The spatial
correlation range of ~875 μm (350 pixels) in the z-direction is associated with millimetre-scale layering, in
contrast to the range of ~150 μm (60 pixels) in the x- and y-directions (Fig. 9 d, e) associated with the size
of a grain.
To investigate the nature of the variability in the z-direction, a larger sub-sample (7145 x 7145 x 9330
μm³) of sandstone S1 was imaged with a resolution of 5 μm (Fig. 10). A slice-by-slice porosity evaluation in
the z-direction (Fig. 10c) shows two cycles with a period of ~3000 μm and a range of ~2000 μm (Fig. 10f).
This is in addition to the finer-scale periodicity at ~150 μm, which was also observed in the x- and y-
directions (Fig. 10d, e). Therefore, to capture this layering, a cube with an edge length of at least ~2000 μm,
which was derived from the directional analysis, should be defined as the REV for S1. Alternatively, the
REV from the classic approach was with an edge length of 475 voxels (~1187 μm, Appendix B, Figs. B1a,
b). As the directional REV approach better captures the anisotropy in the sample structure, the entire
specimen cube with an edge length of 2950 μm (~1.5 larger than the evaluated directional REV) scanned
with a 2.5 μm resolution was chosen for the flow modelling.





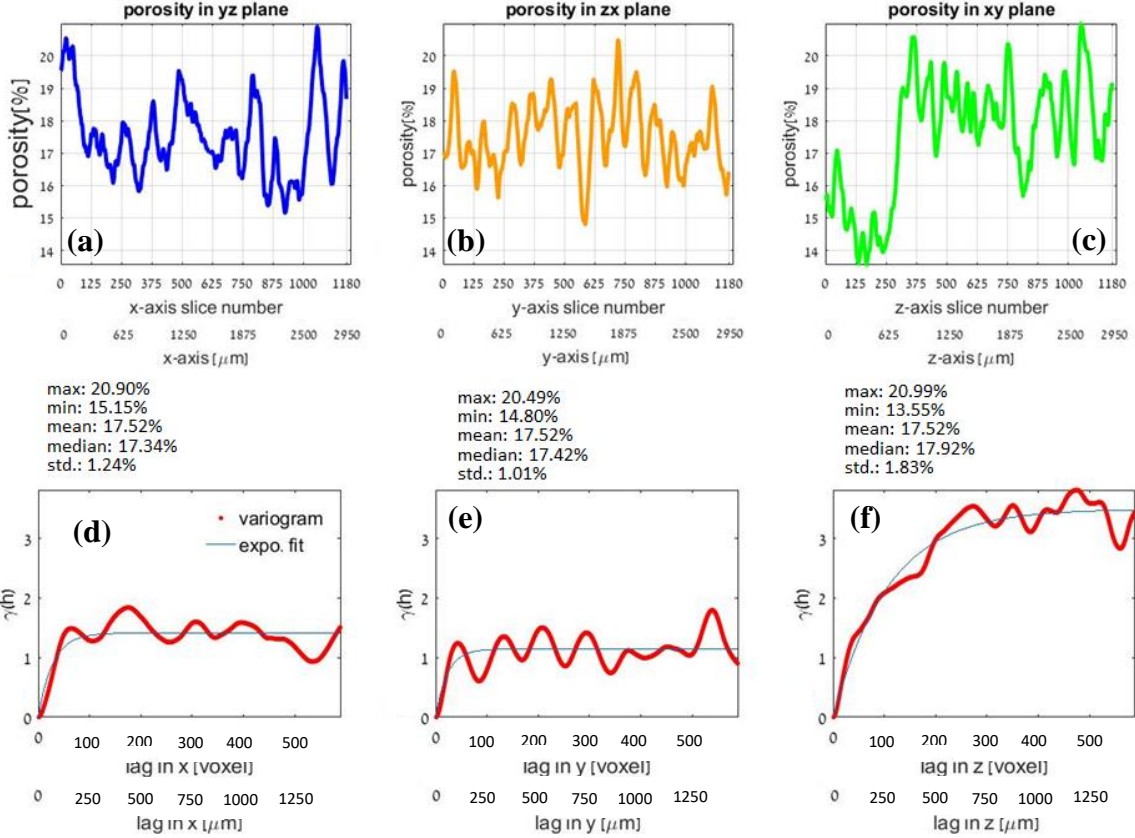



**Figure 9:** *Directional REV analysis of sandstone S1 scanned with a resolution of 2.5 μm. In the top row, the porosities calculated slice by slice for the x-, y- and z-directions are presented (**a-c**). In the bottom row (**d-f**), the results of the variogram analysis are presented. The cyclicity in the variogram refers to the cyclicity of the porosity at the pore scale.*





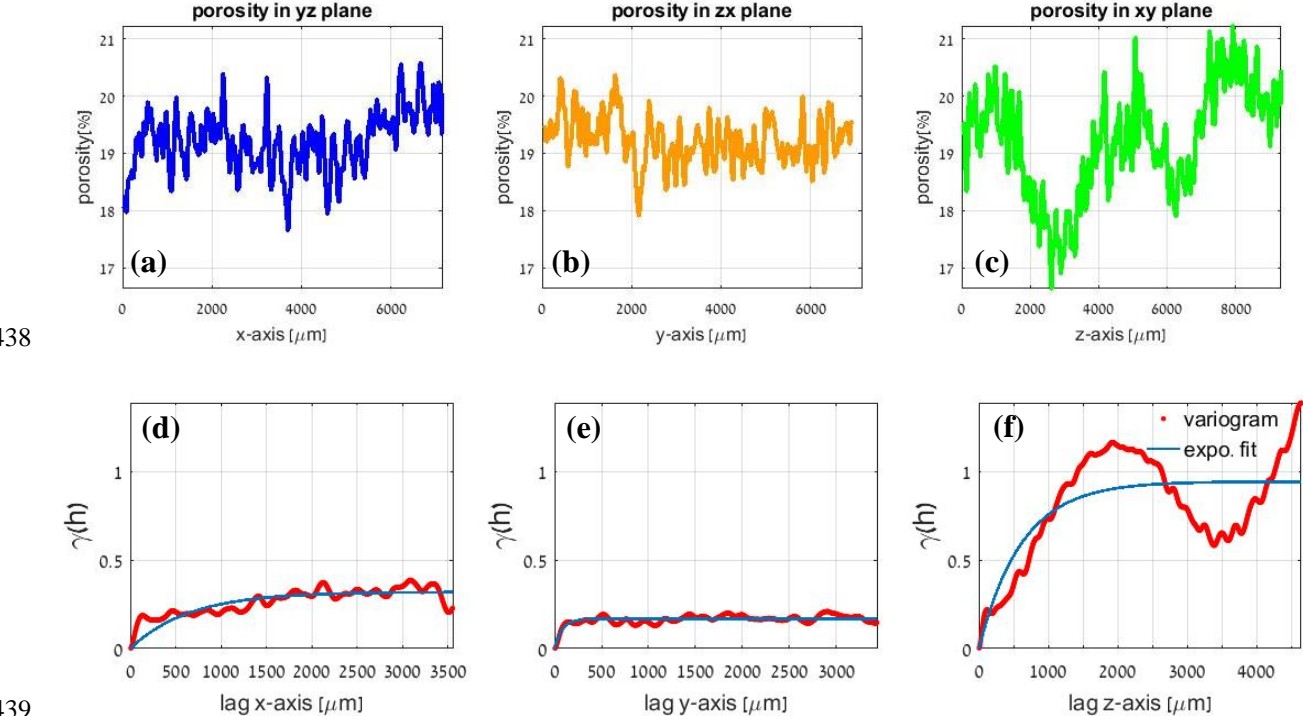

**Figure 10:** *Directional REV analysis for sandstone S1 scanned with a resolution of 5 μm. In the top row, the porosities calculated slice by slice for the x-, y- and z-directions are presented (**a-c**). In the bottom row (**d-e**), the variogram analysis shows cyclicity in the x- and y-directions associated with pore size fluctuations. (**f**) In the z-direction, the range of ~2000 μm is associated with porosity fluctuations between the high- and low-porosity bands separated by this distance.*

Figure 11 shows the results of the directional REV analysis for sample S2 conducted on a cube with an edge length of 2950 μm (1180 pixels) scanned with a 2.5 μm resolution. Each direction shows a remarkably different trend (Fig. 11a-c). The largest difference between the minimum and maximum slice porosities, 6.57 %, appears in the z- direction (in contrast to 4.5 % and 3.56 % in the x- and y-directions, respectively), and the standard deviation in the z-direction (1.53 %) is approximately twice those in the other two directions (0.86 % and 0.73 %). An increase in porosity with the slice number is observed in the z-direction (Fig. 11c) and is also represented by the trend in the variogram (Fig. 11f). This trend is inversely correlated with the content of clay between the sand grains (Fig. 11a-c, brown curve). The negative correlation coefficient between the porosity and clay matrix in the z-direction (-0.87) (Fig. 11g-i) is larger





(in its absolute value) than the corresponding correlation coefficients in the x- and y-directions (-0.34 and -
0.03, respectively). Finally, the sill in the directional variogram analysis is not reached for S2 in the y- and
z-directions for the cube with an edge length of 2950 μm. Alternatively, the large difference between the
mean and median porosities from the classic REV approach (Appendix B, Fig. B1c, d) implies that the REV
also cannot be reached. As a result, flow modelling could not be conducted for sample S2.

**Figure 11:** *Directional REV analysis for sandstone S2. In the top row, the porosities calculated slice by slice*
*for the x-, y- and z-directions and the fractions of the clay matrix are presented (**a-c**). In the middle row (**d-f**),*



*variogram analyses indicate that the sill is not reached for the y- and z-directions. Scatterplots (**g-i**) show the*
*correlation of the porosity with the clay content, revealing a prominent inverse trend in the z-direction.*

Directional REV analysis conducted on the cube of sample S3 with an edge length of 2950 μm (1180 pixels) scanned with a 2.5 μm resolution indicates that all three directions show similar fluctuations around the mean porosity (Fig. 12a-c). The differences between the minimal and maximal IPs are 5.89 % in the y-direction and 5.45 % in the x- and z-directions. The standard deviation is similar in all directions. Variogram analysis (Fig. 12d-f) indicates similar sills (1.1-1.3) and ranges (~150 μm (60 pixels)) in all directions, indicating an isotropic pore network. Classic REV analysis (Appendix B, Fig. B1) yields an REV with a cube edge length of 875 μm (350 pixels), which is larger than the edge length of 150 μm from directional REV analysis. Therefore, the result of the classic REV analysis (875 μm or 350 pixels) was used in this study for the flow modelling.





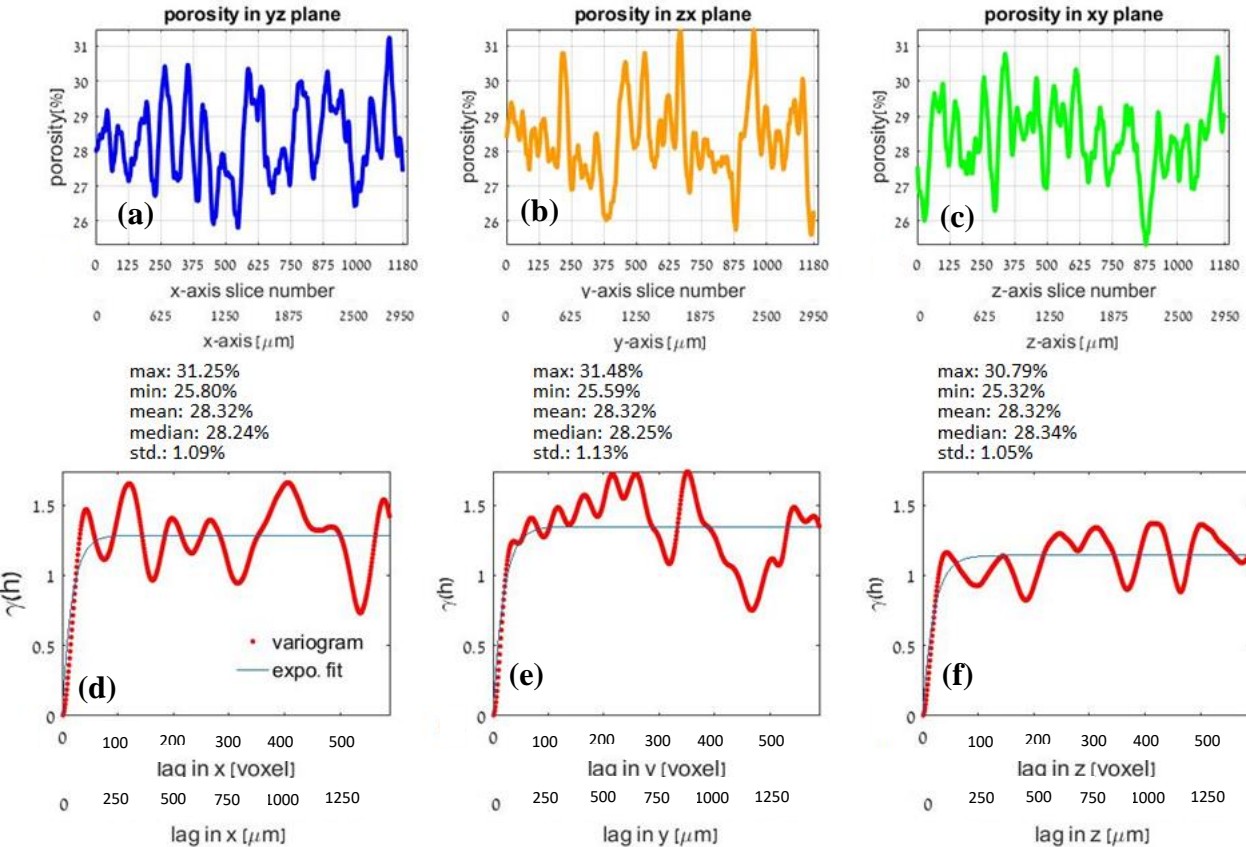

*Figure 12: Directional REV analysis for sandstone S3. In the top row, the porosities calculated slice by slice for the x-, y- and z-directions are presented (a-c). In the bottom row (d-f), the variogram analysis indicates the representativeness of the sample for relatively small sub-volumes with a cube edge length of ~150 µm (60 pixels).*

**4.3. Fluid flow modelling at the pore scale**

Fluid flow was modelled at the pore scale in two different micro-CT-scanned geometries: 1) a full cube of sample S1, including two adjacent parts possessing relatively low (0-250 voxels) and high (250-1180 voxels) porosities (Fig. 9c), and 2) sample S3 within its REV dimensions (Table 3). Modelling of the 3D geometry of sample S2 was not performed due to the reasons detailed above. A constant pressure gradient of $2.424 \left[\frac{Pa}{mm}\right]$ between the inlet and outlet boundaries was applied in all the simulations for consistency.






***Table 3.*** *Porosity losses in S1 and S3 over the course of applying the extended computational workflow (Fig.*
*2).*

| Sample | Sample size (mean mesh edge size) [μm] | CT segmented image porosity (%) | Connected porosity (%) | Mesh porosity (%) | Gas porosity (%) |
|---|---|---|---|---|---|
| S1 (entire sample, 1180 voxels) | 2950 (14) | 17.5 | 15.6 | 13.6 | 28 |
| S3 (REV, 350 voxels) | 875 (5) | 28.3 | 27.9 | 25.9 | 31 |


The porosity of the meshed domain of sample S1 is 13.6 % (in contrast to 17.5 % in the segmented
image, Table 3), and the mesh edge length is 14 µm along the pore walls. The observed porosity loss results
from disconnecting narrow pore throats from the connected cluster imaged with a 2.5 µm voxel size due to
the use of a 14 µm mesh size (the lowest possible for our computational needs). A maximum Reynolds
number of $Re = 0.084$ was used to guarantee the simulation in a creeping flow regime.
The symmetrized permeability tensor, $\bar{\bar{\kappa}}$, was obtained as follows (Table 2):
$$\bar{\bar{\kappa}}_{sym} = \begin{pmatrix} 420 & 66.3 & 1.91 \\ 66.3 & 344 & 12.8 \\ 1.91 & 12.8 & 163 \end{pmatrix}$$
The permeability tensor is anisotropic, with $\kappa_{zz}$ being more than half $\kappa_{xx}$ and $\kappa_{yy}$. This result is in
agreement with the appearance of horizontal banding with higher cementation derived from the variogram
analysis (Fig. 10f).
The porosity of the meshed domain of sample S3 is 25.9 % (in contrast to 28.3 % in the segmented
image, Table 3), and the mesh edge length is 5 µm along the pore walls. A maximum Reynolds number of
$Re = 0.22$ was used to guarantee the simulation in a creeping flow regime. The symmetrized permeability
tensor is close to isotropic (Table 2):
$$\bar{\bar{\kappa}}_{sym} = \begin{pmatrix} 4517 & 5 & 38 \\ 5 & 4808 & 547 \\ 38 & 547 & 4085 \end{pmatrix}$$





The tortuosity of S3 in the x-, y-, and z- directions varies in the range [1.39, 1.47] (Table 2), and the
largest value is observed in the z-direction, which is in agreement with the lowest permeability in the z-
direction.
**4.4. Image analysis**
For S1, the mode peak of the pore size distribution (measured by a Feret maximum calliper) (Fig. 13,
red line) is at 194 µm (Table 2). In total, 3500 pores were analysed. The pore specific surface area (*PSA*)
calculated from micro-CT images is $0.068\,\mu m^{-1}$. The tortuosity, measured from the whole CT image,
indicates similar values in the x- and y-directions of 1.37 and 1.38, respectively, whereas in the z-direction,
the tortuosity is 1.48 (Table 2). As many paths were considered, we suggest that this difference is created by
the textural features that appear in horizontal planes (Fig. 3a).
For S2, the mode peak of the pore size distribution (Fig. 13, blue line) is at 21 µm. A large pore
population is also recognized at ~100 µm (Table 2). In total, 45000 pores were analysed. The pore specific
surface      area      (*PSA*)      calculated      from      micro-CT      images      is
$0.136\,\mu m^{-1}$ (Table 2), which is twice as large as the *PSA* of S1.
For S3, the pore size distribution (Fig. 13, green line) has a mode peak at 223 µm and shows a
Gaussian shape (Table 2). In total, 3491 pores were analysed. The geometry-based tortuosity values
measured from the whole CT image with multiple paths is 1.32, 1.34 and 1.39 in the x-, y- and z-directions,
respectively. The tortuosity is lower for S3 than for S1 in all directions, which is a direct result of the smaller
amount of cement in the pore throats. The *PSA* of S3 is $0.069\,\mu m^{-1}$, which is similar to that of S1.





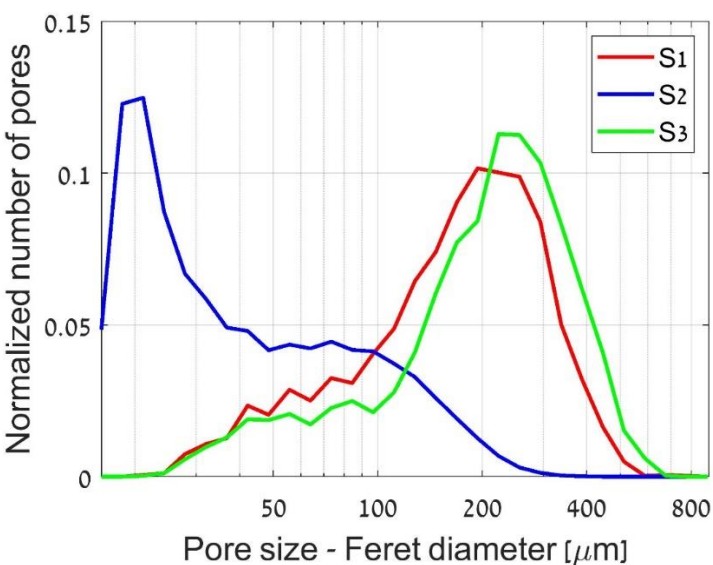


*Figure 13: Statistics of the pore sizes calculated by image analysis for three sandstone samples: S1, S2, and*
*S3.*




**5.    Discussion**

**5.1. Validation of permeability by micro- and macro-scale rock descriptors**

Each of the evaluated micro- and macro-scale rock descriptors supplies qualitative information about

the sample permeability (Tables 2-3), which is used to validate the multi-methodological approach presented
in this paper. Specifically, the increasing mercury effective saturation with increasing pressure shows a
similar *PTSD* curve slope for sandstone samples S1 and S3 in the macro-pore throat range (Fig. 7),
suggesting that these samples have similar structural connectivity. However, S1 has a smaller volume
fraction of pore space available for fluid flow that is controlled by macro pore throats (i.e., 81 % in S1 vs. 93
% in S3, Fig. 7) due to its higher contents of silt, clay, and Fe-ox cement. The intermediate layer (S2)
comprises more fines, which form a clay matrix with 19 % porosity, most likely affected by the burial
conditions of this sample (Table 2). Only ~15 % of the pore volume fraction in S2 is controlled by bottle-
neck macro pore throats (Fig. 7). However, the characteristic length of S2, 12.3 µm (Table 2), indicates that




macro-pore connectivity is still possible even when the pore space consists mainly of sub-macro-scale porosity. This 0.15 volume fraction is in agreement with Harter (2005), who estimated a volume fraction threshold of 0.13 for correlated yet random 3D fields required for full interconnectivity.

The value of the connectivity index of S3 (10) is approximately three times higher than that of S1 (3.49), while both rocks are defined as moderately sorted sandstones (Table 2). This difference is due to S1 having a smaller number of inequivalent loops within the pore network than S3 (Appendix C), leading to smaller $\beta_1$ values in Euler characteristics. Inequivalent loops are correlated with pore throats; their number is affected by the resolution of the CT image and by the partial volume effect at grain surfaces (Cnudde and Boone, 2013; Kerckhofs et al., 2008), where some voxels could be identified as grains and thus "clog" the small pore throats. Artefact porosity loss is apparent for S1, where the IP is 17.5 % (in contrast to the CT porosity of 23.5 % predicted from MIP, Table 2). The connectivity index of S2 (0.94, Table 2) is lower than those of both S1 and S3 because of the clay matrix, which clogs pores. The effect of the partial volume effect on the image connectivity and on the preservation of small features was reviewed by Schlüter et al. (2014).

A correlation was found between the grain size and the amount of Fe-ox cement in S1 evaluated at each slice along the z-direction (from the image analysis, Fig. 14). Exceptionally large grains are detected (indicated by the red rectangle) near the cemented region at ~750 μm. Large grains and a relatively high amount of cement can also be observed in the S1 thin section (Fig. 3b). Large grains cause large pores and generate relatively permeable horizons through which water flow and solute transport can become focused (McKay et al., 1995; Clavaud et al., 2008), supplying iron solutes. We suggest that a vadose zone was formed after flooding events, where the water flow mechanism could have changed from gravity dominated to capillary dominated. Water then flowed due to capillary forces along grain surfaces towards regions with larger surface areas, and iron solutes precipitated in a reaction with oxygen available in the partly saturated zone. We suggest that with time, this cementation mechanism caused a decrease in the pore throat size near the preferential path, while the preferential path with a low surface area remained open, eventually generating the observed anisotropic flow pattern.





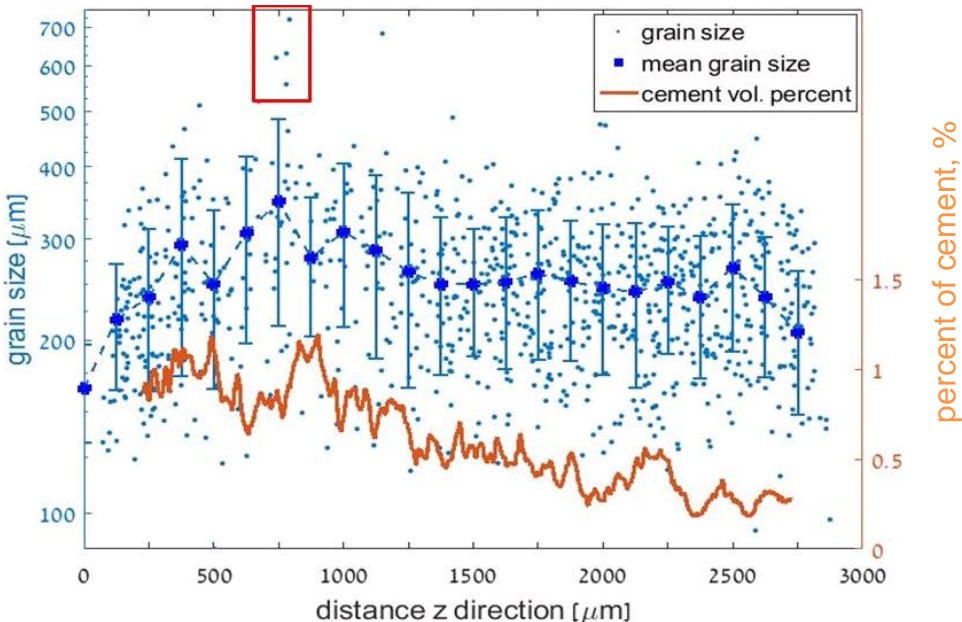


**Figure 14.** *Grain size scattering and Fe-ox cement content in sandstone S1 in slices along the z-direction.*

In this respect, permeability anisotropy in sandstones at a small scale is usually attributed to the shape
or preferential orientation of grains and pores (e.g., Sato et al., 2019) and to a heterogeneous distribution of
cementing material at grain contacts (Louis et al., 2005). At a larger scale, a higher degree of permeability
anisotropy is associated with the presence of localized beds, foliation, and compaction bands that constitute
barriers to flow in the perpendicular direction (see Halisch et al., 2009; Clavaud et al., 2018 and references
therein). Clay-free and cement-free layers constitute the main avenues for flow in the parallel direction (e.g.,
Fig. 14).
The directional REV analysis of S1 reveals porosity bands with a period of 2 mm perpendicular to the
z-direction (Fig. 10c, f) that are composed of sub-domains exhibiting high and low Fe-ox cementation. Flow
modelling in the specified REV shows anisotropy (Table 2) and an average permeability value of 310 mD
that is close to that derived from MIP (330 mD). However, the average permeability is lower than the
average experimental gas permeability (~543 mD); this difference should be related to the loss of porosity
due to limitations on the CT resolution, image processing and meshing (Table 3, see Sect. 5.2 for more
details).





In contrast, no banding was detected in S3 by the directional REV analysis (Fig. 12). Flow modelling
and upscaling to the macro scale indicate an isotropic sample (Table 2), which agrees with the isotropy
derived from the variogram analysis. However, the modelled permeability (~4500 mD) is ten times higher
than the MIP-derived permeability (466 mD, Table 2). Gas permeability measurements indicate anisotropy,
yielding permeabilities of 4600 mD in the x-y plane and 220 mD in the z-direction (with an anisotropy ratio
of ~20, defined here as $\kappa_{\parallel}/ \kappa_{\perp}$, e.g., Tiab and Donaldson, 2004). For comparison, the values of this ratio
obtained from experimental permeability measurements were ~1.2 for Bentheim sandstone (Louis et al.,
2005), ~1.7-2.5 for a sandstone within the Cretaceous Virgelle Member, Alberta, Canada (Meyer and
Krause, 2001), and ~8.5 for Berea sandstone (Sato et al., 2019). However, in some laboratory measurements
conducted parallel to the layering (in the x-y plane), poorly cemented grains in S3 were dislocated from the
weakly consolidated sample due to the application of a pressure gradient. This could have resulted in a
higher measured gas flux and thus a higher permeability parallel to the layering, yielding a high anisotropy.
In this case, the permeability upscaled from the modelling in S3 is also exaggerated.
Alternatively, the disagreement between the laboratory-determined permeability (perpendicular to the
layering) and the permeability obtained from the flow modelling (Table 2) may also stem from the small
dimension of the modelled domain (cube edge length of ~0.875 mm), which may not have included the
textural features that constrain fluid flow on a larger scale (e.g., Fig. 5d).
For sample S2, both REV analyses indicated an REV size much larger than the investigated sample
size (Fig. 11, Figs. B1 c,d in Appendix B). For this reason, the analytical programme formulated in our
paper cannot entirely be applied to S2 due to the impossibility of determining a reliable REV and hence
conducting pore-scale flow modelling. As a result, although sample S2 represents a common sandstone, it is
of a very heterogeneous nature, and a sample larger than 3 mm is required to capture its REV. The MIP-
derived permeability is 4 mD; this low permeability is due to a clay-rich matrix that encloses substantial
void space (Hurst and Nadeau, 1995; Neuzil, 2019). The gas permeability of the quartz wacke layer (S2,
~4.6 mD on average) is approximately 2 orders of magnitude lower than that of the quartz arenite layers (S1
and S3, Table 2). The permeability anisotropy ratio in S2 is ~2.8. The high inverse correlation between the
porosity and clay matrix content enhanced in the z-direction (Fig. 11 g-i) suggests that the clay matrix
pattern appears as horizontal layering, thus generating the observed anisotropy.






**5.2. Upscaling permeability: accuracy of the extended computational workflow**


The extended computational workflow (Fig. 2) serves as the main tool in this study for upscaling
permeability from the pore-scale velocity field. The accuracy of each step in the workflow affects the
ultimate result.

Following the steps of the workflow (Fig. 2), a micro-CT image resolution of 2.5 μm limits the
reliability of the representation of the porous medium and defines the lower pore identification limit using
this method. As an example of this limitation, the *SSA* (bulk specific surface area) calculated by MIP is
larger than the *PSA* (pore specific surface area) calculated by micro-CT image analysis in all the samples
(Table 2), although the pore volume is always smaller than the bulk volume. The *PSA* from micro-CT is
limited by the image resolution and therefore does not consider relatively small pores with large surfaces.
The *PSA*s of S1 and S3 are similar, but the *SSA* (from MIP) of S1 is 20 times larger than that of S3 because
S1 has a larger surface area at small pores created mainly by Fe-ox cement (compare Fig. 3c-f for S1 to Fig.
5c for S3). S2 shows a *PSA* twice as large as that of S1 due to the presence of clay and clay matrix with
large surface areas.

Image processing and segmentation were applied in this study to recover the image geometry, which
was blurred by noise or affected by the partial volume effect (see Sect. 3). Then, the loss of pore space due
to the resolution limits was estimated in this study from the amount of mercury filling the pores with
diameters equal to the resolution limit (Fig. 7a). After segmentation, sample S1 had a segmented image
porosity of 17.5 % and a CT predicted porosity of 23.5 % from MIP (Tables 2, 3). Therefore, the difference
in porosities generated by the partial volume effect in the image processing scheme (e.g., Cnudde and
Boone, 2013) is a significant component of error, especially for small structures, such as pores with a large
surface area-to-volume ratio. In contrast, the image porosity of S3 after segmentation was 28.3 %, which is
close to the porosity of 30.4 % estimated from MIP (Tables 2, 3). This is a result of the very small degree of
cementation and the absence of Fe-ox flakes in the majority of the sample pores, leading to the small
contribution of the partial volume effect. In comparison, a fine-grained and well-sorted Lower Cretaceous
Fm. sandstone from Heletz Field (e.g., Fig. 1a) (Tatomir et al., 2016) comprising clay and calcite had MIP
and micro-CT porosities of 26.7 % and 20.9 %, respectively.





An additional source of inaccuracy is the use of a porosity-based REV for permeability
approximations. Mostaghimi et al. (2013) showed that for CT images of sandpacks (homogenous samples),
the porosity-based REV had an edge length of 0.5 mm, whereas the permeability-based REV was twice as
large. Moreover, the porosity- and permeability-based REVs in images of bead packs derived by Zhang et
al. (2000) had edge lengths of 1.71 and 2.57 mm, respectively. According to Mostaghimi et al. (2013), larger
REV values for permeability rely on contributions from the tortuosity and connectivity of pore spaces,
whereas the larger REV values of Zhang et al. (2000) may be related to the heterogeneity of the sample.
Further, textural bedding at a 2 mm scale dominates the porosity anisotropy in S1 (Fig. 10f, evaluated
by the directional REV, e.g., Halisch, 2013). To upscale to permeability reliably, the REV domain should be
sufficiently large such that it is bounded from below by the scale of the textural bedding (i.e., an edge length
> 2 mm) but should not be larger than necessary to optimize computational efficiency (while remaining
within the same scale of heterogeneity, i.e., at the macro scale). As a result, a REV consisting of an edge
length of ~2950 μm (~1.5 times larger than the scale of textural bedding) was chosen in the current study in
sample S1. For comparison, in other studies, the edge lengths of REVs in sandstones were 0.68 mm (Ovaysi
and Piri, 2010), 0.8 mm (Mostaghimi et al., 2013), and 1.2 mm (Okabe and Oseto, 2006; Tatomir et al.,
2016). The larger REV size in the current study found by the directional approach (rather than by the classic
isotropic approach) was due to the textural features revealed in the z-direction.
Another source of inaccuracy is the geometry used for the flow model. The geometry considered in
this study included only the pore network connecting six faces of the REV cube. Other pore spaces in the
REV disconnected from the main network were deleted (because all paths smaller than the resolution were
prescribed as grain pixels due to the partial volume effect), thus resulting in the smaller effective size of the
simulation domain. The image porosity of sample S1 was 17.5 %, whereas its connected porosity was
estimated as 15.6 % (Table 3), while those of sample S3 were 28.3 % and 27.9 %, respectively.
Furthermore, the mesh was generated by taking a trade-off between the size of the mesh elements (4
elements in the smallest pore throat) and computational limits into account while coarsening the mesh
elements towards the pore centre. The connectivity between pores with very fine pore throats that could not
be replaced by mesh elements could be lost, resulting in the loss of those pores in the calculations. In sample
S1, the porosity used in the simulation was approximately 50 % smaller than the porosity estimated by gas





porosimetry (Tables 2, 3). In contrast, the porosity used to simulate S3 was mostly preserved, comprising
~84 % of that estimated in the laboratory.

For comparison, in the fine-grained sample of the Lower Cretaceous sandstone from Heletz Field in

Israel (Fig. 1a), which has grain size characteristics similar to those of S1 but with higher clay and additional
calcite contents (Tatomir et al., 2016), the permeability upscaled from micro-CT flow modelling (conducted
by the same simulation method as that in the current study) exceeded the gas permeability by a factor of ~6.
This could be related either to the small REV for the flow model or to the reduction in the specific surface
area by image processing and meshing (Mostaghimi et al., 2013) for the flow modelling.

Finally, the upscaling process from the flow modelling successfully predicted the permeability

anisotropy ratio of ~ 2.3 in S1, as discussed above. For comparison, the permeability anisotropy ratio
evaluated using micro-CT flow monitoring in clay-free sandstones (Clavaud et al., 2008) had a mean value
of ~2.5 (ranging from ~1.7 to ~5.2), related to the presence of less permeable silty layers. This is consistent
with the ratio estimated at the pore scale in Rothbach sandstone (~5) (Louis et al., 2005), attributed to
lamination due to differences both in the characteristics of the solid phase (grain size and packing) and in the
content of Fe oxides.

## 6.  Conclusions

This paper presents a detailed description and evaluation of a multi-methodological petrophysical

approach for the comprehensive multiscale characterization of reservoir sandstones. The validation was
performed on samples from three different consecutive layers of Lower Cretaceous sandstone in northern
Israel. The following conclusions can be drawn:

1.  The suggested methodology enables the identification of links between Darcy-scale

permeability and an extensive set of geometrical, textural and topological rock descriptors,

which are quantified at the pore scale by deterministic and statistical methods. Specifically,

micro-scale geometrical rock descriptors (grain and pore size distributions, pore throat size,

characteristic length, pore throat length of maximal conductance, specific surface area, and





connectivity index) and macro-scale petrophysical properties (porosity and tortuosity), along

with quantified anisotropy, are used to predict the permeability of the studied layers.

2.  Laboratory porosity and permeability measurements conducted on centimetre-scale samples

show less variability for the quartz arenite (top and bottom) layers and more variability for the

quartz wacke (intermediate) layer. The magnitudes of this variability in the samples are

correlated with the dimensions of their representative volumes and anisotropy, both of which

are evaluated within the micro-CT-imaged 3D pore geometry. This variability is associated

with clay and cementation patterns in the layers and is quantified in this study with image

analysis.

3.  Two different scales of porosity variations are revealed in the top layer by statistical variogram

analysis: fluctuations at 150 μm are due to variability in grain and pore sizes, and those at 2 mm

are due to the occurrence of high- and low-porosity bands occluded by Fe-ox cementation. The

latter millimetre-scale variability is found to control the macroscopic rock permeability

measured in the laboratory. Bands of lower porosity could be generated by Fe-ox cementation

in regions with higher surface areas adjacent to preferential fluid flow paths.

4.  The macroscopic permeability upscaled from the pore-scale velocity field simulated by flow

modelling in the micro-CT-scanned geometry of millimetre-scale sample shows agreement

with laboratory petrophysical estimates obtained for centimetre-scale samples for the quartz

arenite (top) layer. The anisotropy in both estimates correlates with the presence of millimetre-

scale bedding, also recognized by the variogram analysis.

5.  The multi-methodological petrophysical approach detailed and evaluated in this paper is

particularly applicable for the detection of anisotropy at various rock scales and for the

identification of its origin. Moreover, this method allows the accurate petrophysical

characterization of reservoir sandstones with broad ranges of textural and topological features.


**Acknowledgements**

This project was supported by fellowships from the Ministry of Energy, Israel, and the University of

Haifa. The authors are grateful to Igor Bogdanov from the University of Pau for his continuing scientific
support. Special thanks to Rudy Swennen and his group from KU Leuven for their contributions to the MIP,



thin section preparation, microscopy and micro-CT image processing; to Veerle Cnudde and her group from
Ghent University for teaching us the image processing techniques; to Kirill Gerke and Timofey Sizonenko
from the Russian Academy of Sciences for providing their image processing code; to Uzi Saltzman from
Engineering Geology and Rock Mechanics Company, Israel, for sending his detailed historic geological
description of the study area; and to Or Bialik, Nimer Taha and Ovie Emmanuel Eruteya from the
University of Haifa, Israel, for their assistance in the laboratory work.

**Competing interests**
The authors declare that they have no conflicts of interest.

**Author contributions.** PH and RK designed the study. PH developed codes for pore-scale modelling with
contributions by RK and MH. BS advised the microscopy and led the geological interpretations. MH scanned
the samples and contributed to the statistical analysis conducted by PH. NW led the laboratory
measurements. All co-authors participated in the analysis of the results. PH wrote the text with contributions
from all co-authors. All co-authors contributed to the discussion and approved the paper.



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



## Appendix A: Description of the Hatira Formation

The Hatira Fm. is the lower part of the Kurnub Group of Lower Cretaceous (Neocomian – Barremian) age. The Hatira Fm. nomenclature used in Israel and Jordan is equivalent to Grès de Base in Lebanon (Massaad, 1976). This formation occurs in Israel in outcrops from the Eilat area along the rift valley, in the central Negev, and in the northernmost outcrops on Mount Hermon; it forms part of a large Palaeozoic – Mesozoic platform and continental margin deposits in northeastern Africa and Arabia. The Hatira Fm. consists of siliciclastic units, typically dominated by quartz-rich sandstones (Kolodner et al., 2009 and references therein). The underlying Palaeozoic sandstones cover large areas in North Africa and Arabia from Morocco to Oman; these sandstones overlie a Precambrian basement affected by Neoproterozoic (pan African) orogenesis (Garfunkel, 1988, 1999; Avigad et al., 2003, 2005). The lower Palaeozoic sandstones in Israel and Jordan originated from the erosion of that Neoproterozoic basement, the Arabian-Nubian Shield, with contributions from older sources. These lower Palaeozoic sandstones (Cambrian and Ordovician) are described as first-cycle sediments (Weissbrod and Nachmias, 1986; Amireh, 1997; Avigad et al., 2005). Exposures of the Hatira Fm. in the Central Negev, the Arava Valley, Eilat and Sinai were originally defined as the Wadi (Kurnub) Hatira Sandstone (Shaw, 1947). The largely siliciclastic section of the Hatira Fm. is intercalated with carbonates and shales representing marine ingressions that increase towards the north (Weissbrod, 2002).

The Lower Cretaceous sandstones of the Kurnub Group are described as super mature, cross-bedded, medium- to fine-grained, moderately sorted to well-sorted quartz arenites with a high zircon-tourmaline-rutil (ZTR) index (for more details, see Kolodner (2009)). Earlier observations indicate the relatively scarce occurrence of siltstones and claystones compared to sandstones (Massaad, 1976; Abed, 1982; Amireh, 1997). These Lower Cretaceous sandstones are mainly the recycled products of older siliciclastic rocks throughout the Phanerozoic; the sand was first eroded from the surface of the pan African orogeny ca. 400 Ma prior to its deposition in the Lower Cretaceous sediments (Kolodner et al., 2009).

The Mount Hermon block was located at the southern border of the Tethys Ocean during the Early Cretaceous (Bachman and Hirsch, 2006). A paleo-geographical reconstruction indicates that the sandy Hatira Fm. (Fig. 1) was deposited in a large basin, which included both terrestrial and coastal environments such as swamps and lagoons (Sneh and Weinberger, 2003). The Hermon block, located next to the Dead Sea





Transform, was rapidly uplifted during the Neogene (Shimron, 1998). The area is marked by intense
erosion, which resulted in extensive outcrops such as those near Ein Kinya on the southeastern side of Wadi
E'Shatr.

**Appendix B: Results of the REV determination by the classical approach**





***Figure B1.*** *Results of the classic REV analysis for sandstones S1-S3 (**a,c,e**). (**b, d, f**) Magnified views of the*
*mean and median porosity trends of S1-S3 calculated for varying edge lengths. The scattering of porosity*
*measured for each sub-volume is shown in blue dots. The laboratory porosities measured by gas*
*porosimetry are shown by a pink line. The image porosity for CT, which was predicted by MIP for the*
*resolution limit, is shown by a yellow line. The mean and median porosity are depicted by red and green*
*lines, respectively.*

**Appendix C: Euler characteristic**

The Euler characteristic is a number that describes the structure of a topological space. The most
intuitive way to think about the Euler characteristic is in terms of its Betti numbers ($\beta_i$):
$\chi = \beta_0 - \beta_1 + \beta_2$

For a 3D object, $\beta_0$ is the number of components, $\beta_1$ is the number of inequivalent loops and $\beta_2$ is the
number of cavities (enclosed voids). In describing the topology of the pore space of a porous rock, it can be
assumed that the solid matrix is connected such that $\beta_2 = 0$. In this case, the Euler number reduces to the
difference between the number of discrete components and the number of inequivalent loops. If all pore
spaces are connected via one pathway or another and assuming that there are no isolated pore spaces, then $\beta_0$
= 1. In a pore network of sandstone that can be modeled as a bundle of tubes, the number of loops $\beta_1$ is large,
and $\chi$ is negative. Therefore, the Euler number, $\chi$, is related to the connectivity of the pore space. As the
number of loops decreases, the Euler number becomes less negative and eventually becomes positive, where
the system will no longer percolate, according to Vogel (2002).





## Appendix D: Maximal hydraulic conductance


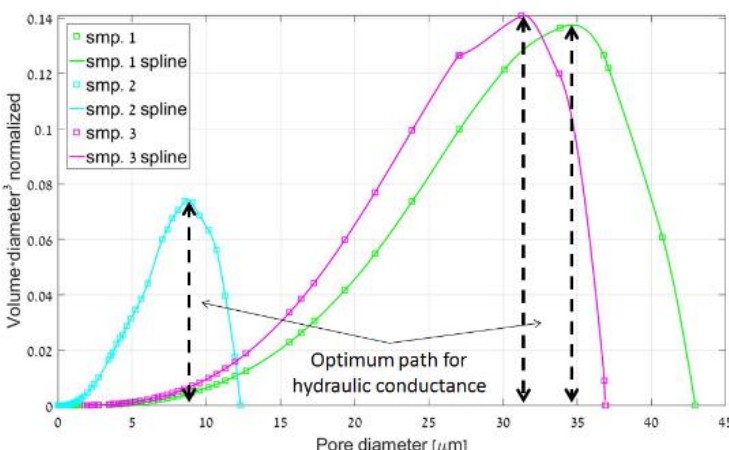


***Figure D1.*** *The pore throat length of the maximal hydraulic conductance, $l_{max}$, is defined from the maximal*
*(normalized) hydraulic conductance (Katz and Thompson, 1987), specified at the vertical axis of the chart.*
*The corresponding pore throat diameters (x-axis) marked by black arrows define the pore throat diameters*
*(or pore throat lengths of maximal conductance), $l_{max}$, where all connected paths composed of $l \geq l_{max}$*
*contribute significantly to the hydraulic conductance (see Sect.3.2).*