# Peer review of "Benchmark study using a multi-scale, multi-methodological approach for the"

_Solid Earth, 2020_

## Referee Comment (RC1) · Anonymous Referee #1 · 2 Aug 2020

**1    General comments**

I read the paper of "Benchmark study using a multi-scale, multi-methodological approach for the petrophysical characterization of reservoir sandstones" with great interest. It deals with so called "digital petrophysics", a characterization of petrophysical properties of rocks, essentially but not limited to porosity and permeability, by evaluation of 3D microCT scans through different statistical and analytical techniques mutuated from image analysis. In this particular study, 3 sandstone samples from different layers of the Hatira formation in northern Israel were characterised.

The samples were subjected to a good array of analyses, from more standard such as gas and mercure porosimetry, to more advanced ones such as indeed 3D imaging by microtomographic scanning and the subsequent analysis and flow modelling, in the ultimate quest to obtain in silico reliable "macroscopic" (Darcy scale) description from the pore space imaging. The authors detail their characterization of the spatial variability and variographical analysis of the 3 microCT samples in order to obtain the minimum size of REV, which they claimed being only possible for two of the three samples. Once the REV size determined, Navier-Stokes simulations on the discretized pore space were run to compute the full apparent permeability tensor of the samples. The discussion puts their findings in perspective with the current state of knowledge in this matter, notably enumerating possible sources of errors.

Up to a few imprecisions in the description of the employed geostatistical instruments, the paper has a good scientific quality and practical relevance, and it is written in good english. Even though no groundbreaking results or new methods are introduced, it is the combined use of different methods the strength of this paper, which I believe merits publication in SE after some improvements.

Before diving into the specific and technical comments, a remark: since the paper claims to be a "Benchmark study", I strongly advocate to make at least the segmented 3D scans available in one of the many available public data repositories.

**2  Specific comments**

1. Please use "semivariogram" instead of "variogram", or state in the text that you are using "variogram" to intend "semivariogram", as I will do from now on in this review.

2. Is the support used to compute the experimental variograms the whole image plane orthogonal to the computed direction? It is not completely clear to me

at the moment, I find the Methods description misleading in this regard. If it's the case, I strongly advice to use for variography at least squares and not the full slices, and even better yet would be to do a full 3D variography, doing cubic supports as you did in the "classical" REV analysis of figure B1. Though, this is much more computationally intensive and may require ad-hoc coding. This may have a potentially large effect on the results, depending on the chosen "box side", especially for the sample S2, and if done properly, also actually hint at the true principal axes of anisotropy for the samples, which may not be aligned with their sides.

3. Fig 9-11: if you are showing the fitted variograms, based upon which you define the apparent ranges, I believe you should also state which variogram model was fitted. From the legend I may assume it's an exponential model but or spherical or something else, and the "expo. fit". If it's the case, then you need to specify if the reported "range" is the actual coefficient in the exponential model or the "practical range" of the asymptotic function.

4. All three samples represents exemplary cases for zonal anisotropy, where the sills of the variograms are not constant following different directions. This reinforce my suggestion of making the samples available to the public.

5. It is to me however striking - and this may hint to a too large support definition, cfr comment 2., or else to a graphical imprecision - that no experimental variogram displays any nugget effect. This could mean that the variable has been excessively regularised. Please state in the text how the lags for the calculation of the experimental variagrams were chosen, and if the computed pairs at each lag bin are comparable.

6. Fig 11. Regarding the variograms of sample S2, they are clearly linear, especially the xy plane, which is a clear sign of non-stationarity, as also clear from the strong trend in subfigure (c). However you also correctly recognised the "external drift"

represented by the clay content. This is possibly a textbook example of external drift, which makes the de-trending of porosity worth. My point here is, that the fact that the sample is clearly strongly anisotropic and non-stationary does not mean that it is not possible to extract a REV from it, at least for the two other directions, but with some manipulation, also in the xy plane. Moreover, a full 3D variography (if my 2. comment is valid) may give different insights and results.

7. No histogram of apparent image porosity is displayed, neither from the slices used for the variographic REV nor from the subsets of figure B1, although from that figure we get an idea of the "density" (however there is sampling involved here, I assume). Is it possible that the "cube" porosity - at a given cube size - is also lognormally distributed? Possibly then it could be worth to perform the variographic analysis on a log porosity.

8. For sample S3 the REV is identified at 350 voxels, though only one permeability simulation is conducted. It would be nice to demonstrate that the calculated permeability is somewhat "continuous" by repeating the flow simulations on different subsets of that size of the original microCT image.

All other parts of the paper are satisfactory for me.

**3   Technical comments**

There are a very few typos in the text but given the necessity of more indepth revision from the authors I believe it's not useful to point them out now.

---

## Referee Comment (RC2) · Anonymous Referee #2 · 11 Aug 2020

General Comments

This paper has made commendable efforts to using a multi-scale, multi-methodological approach for the petrophysical characterization of reservoir sandstones. The strength of the study lies in its multiple datasets generated and used. However, the paper requires improvements before it can be ready for publication.

The main aims/objectives of this study should be made very clear from the start. Are you proposing multi-methodological approach for the petrophysical characterization of reservoir sandstones as the best or only method? Or what exactly are you aiming for?

How the achievement of these aims/objectives contribute to the current knowledge

gaps should be clearly discussed in the relevant section of the paper.

The methods need to be clearly discussed.

Many figures require attention.

I do feel that testing all the proposed conclusions made from this study with sandstones from other places will make these conclusions stronger. If it is possible to have sandstones from other places to test your conclusions, this will be very good. However, if the aims/objectives of the study do not require/permit this, then no problem.

Specific Comments

Title

Why is there 'benchmark' in the title? Depending on the revised aims/objectives of the study, the title may require revision.

Introduction

An extended literature review is required. This may be part of the introduction or may be a separate section. This is important to discuss the state of the subject matter and to present a framework and context for which current study fits into. The current introduction is short while the aim of this study does not seem to address some of the issues raised (lines 66-67) in previous studies.

Geological Setting

Abbreviation in line 98 needs to be written in full at first time.

Fig. 1 needs to be increased in size to make it legible. The quality of 1d needs to be improved. 1 a needs lines of latitude and longitude.

Appendix A. It is a bit strange that important geological information is put in an appendix. The key geological information in the appendix needs to be summarised and placed under geological setting. The information presented in this section is too shallow and only focused on a formation. Every relevant geological information about the whole basin and other formations should be included here to give a very good context for the current study.

Methods

3.1. How many samples were collected? Is it possible to state the size of these samples and large block samples or show their photos so that readers can have an idea of how big/small they are. There needs to be proper descriptions of all these samples: how can a reader identify/differentiate a sample from large block samples and from a sub-sample?

Table 1. 3.7 should be 'Optical microscopy'

3.2 The laboratory methods are not properly discussed and this is not good enough. More than just mentioning the names of equipment used, the procedure needs to be properly discussed or appropriate references provided.

If the methods are properly discussed, I do not see any need for Table 1

Only the relevant information needed to understand the procedure for the workflow method should be provided. The current format appears to be excessive.

Results Line 269, 314, -what heavy minerals?

268, 270…referencing methods using 'according to', 'following' should be amended using journal style

276; Mn-Ox…what is this? Please explain?

317-include reference

318….result is mixed with interpretation. Only the results should be presented in the result section in this place and throughout the manuscript.

Fig 4d-scale is missing

347-349 should be moved to the methods section

Fig 6 and 7, 9, 11, 12 should be increased

480-485: more or less a repetition. Any new information here should be moved to methods section. The results of the modelling should be presented here.

509-511: needs to be in the methods section

513-514: is this result or interpretation?

545: if the information in appendix C is important for the discussion, why is it not included in the main body of the manuscript?

560: gravity-dominated ?, capillary-dominated ?

601: use 'study' instead of paper...here and throughout the manuscript

603: 'very heterogeneous in nature' ?

References

I have not bothered to check the references at this stage. The author needs to ensure that all cited references are in the bibliography and vice. For example, I am not sure if I encountered Akinlotan 2018 in the text but it is in the bibliography. Please look into this and others and ensure referencing is accurate.

---

## Editor Comment (EC1) · Joachim Gottsmann (Editor) · 12 Aug 2020

Dear Authors,

I have now received 2 independent reviews of your manuscript. Both reviews are positive and indicate substantial potential in your work for final publication in SE. Both reviewers agree that your manuscript represents a substantial contribution to the scientific progress on the petrophysical characterisation of sandstone.

That being said, both reviewers request that the ms be revised before I can consider it for my final editorial decision.

[Figure]

Reviewer 1 has made concrete suggestions as to how to improve the paper which I invite to consider in full.

Reviewer 2 has also made concrete suggestions for improvement, but is slightly more critical of the approach as depicted in the current version of the ms. They, for example, request that Appendix A is moved to the main text. I second this request which should not be too arduous a task.

There appears to be the opportunity to place your research in a broader context given the limitations and outstanding issues in rock characterisations mentioned in the introduction. It would help the reader to ensure that your research is placed in direct context to the current shortcomings both in the intro and the discussion.

Figures 9-12 are of poor quality and need your attention. I have a hard time reading the labels and axis descriptions. It is not obvious that some should read "lag" as the body of text is capped at the bottom.

In short, I expect to receive a throughly revised ms with a point-by-point record as to how the recommendations by the reviewers and myself have been dealt with.

With best wishes, J. Gottsmann

Executive Editor SE

---

## Author Comment (AC1) · 27 Aug 2020

**Reviewer #1: Answers / discussion (black) of specific review comments (blue)**

1. *Please use "semivariogram" instead of "variogram", or state in the text that you are using "variogram" to intend "semivariogram", as I will do from now on in this review.*

    Thanks for clarifying that! We will change "variogram" to "semivariogram" throughout the entire manuscript as you suggested.

2. *Is the support used to compute the experimental variograms the whole image plane orthogonal to the computed direction? It is not completely clear to me at the moment. I find the Methods description misleading in this regard. If it's the case, I strongly advice to use for variography at least squares and not the full slices, and even better yet would be to do a full 3D variography, doing cubic supports as you did in the "classical" REV analysis of figure B1. Though, this is much more computationally intensive and may require ad-hoc coding. This may have a potentially large effect on the results, depending on the chosen "box side", especially for the sample S2, and if done properly also actually hint at the true principal axes of anisotropy for the samples, which may not be aligned with their sides.*

    Yes, that is the case, concerning your comment on the support. In the manuscript we used a "segmented image slice" which we will rephrase a little bit in order to point out that we used the "whole segmented image slice" for the analysis.
    Additionally, we understand the importance of performing the different types of a variography, even with cubic supports in 3D. In fact, this would be a very good approach in our opinion, if no information about the rocks of interest is available. For this study, we made large efforts to take the samples in the field as accurate as possible to achieve a sub-sampling procedure in line with the visible foliation of the different layers. By best of our knowledge and experience, the resulting "coordinate system" of the samples as well as of the related 3D images should be in very good alignment to the true principle axis of anisotropy. Evidence for this is given by comparing the exceptionally good results of measured and modelled permeability values and tensors. Accordingly, we have chosen the "less challenging" 2D variography path for our study, since we know all of the rock samples' features and textures very well.
    Nevertheless, we will address this topic within the discussion section of the manuscript, as a different methodical approach with its own advantages and drawbacks, inserting the additional results/charts suggested by the reviewer into the supplementary material. We believe that the application and comparison of results from the different variography methods is a big stand-alone topic. It will be of a great impact and thus deserves to be addressed within a new topical manuscript applicable to samples with various (not known) positions of the principal anisotropy axes (which we are already checking for feasibility).

3. *Fig 9-11: if you are showing the fitted variograms, based upon which you define the apparent ranges, I believe you should also state which variogram model was fitted. From the legend I may assume it's an exponential model but or spherical or something else, and the "expo. fit". If it's the case, then you need to specify if the reported "range" is the actual coefficient in the exponential model or the "practical range" of the asymptotic function.*

    Again thanks for pointing this out. In fact, we used the "practical range" of the asymptotic function as "the range" for the semivariograms. It seemed more "intuitive" to potential users which are not "deeply in this statistical topic". We will clarify that in the text.

4. *All three samples represents exemplary cases for zonal anisotropy, where the sills of the variograms are not constant following different directions. This reinforce my suggestion of making the samples available to the public.*

All data are available at the PANGÄA-repository as explained in the "Supplementary materials section". Related doi's will be added at the end of the manuscript and cited in the text of the manuscript.

5. *It is to me however striking - and this may hint to a too large support definition, cfr comment 2, or else to a graphical imprecision - that no experimental variogram displays any nugget effect. This could mean that the variable has been excessively regularised. Please state in the text how the lags for the calculation of the experimental variograms were chosen, and if the computed pairs at each lag bin are comparable.*

Obviously, the nugget effect can be attributed to measurement errors or spatial sources of variation at distances smaller than the sampling interval or both. Measurement error occurs because of the error inherent in measuring devices. Natural phenomena can vary spatially over a range of scales. Variations at microscales smaller than the sampling distances will appear as part of the nugget effect. Nevertheless, this is more or less impossible to achieve for "mining" spatial data from μ-CT images since we have a fixed resolution limit. Hence, all variation below that "hard resolution boundary" is "invisible" for the variography analysis. This clearly is a drawback of the image analysis, and hence it is important to gain detailed understanding of the scales of spatial variation from multiple methods. Accordingly, the smallest lags are related to the resolution, i.e. the smallest segmented feature of the 3D scan. We will address this issue within the discussion section.

6. *Fig 11. Regarding the variograms of sample S2, they are clearly linear, especially the xy plane, which is a clear sign of non-stationarity, as also clear from the strong trend in subfigure (c). However, you also correctly recognised the "external drift" represented by the clay content. This is possibly a textbook example of external drift, which makes the de-trending of porosity worth. My point here is that the fact that the sample is clearly strongly anisotropic and non-stationary does not mean that it is not possible to extract a REV from it, at least for the two other directions, but with some manipulation, also in the xy plane. Moreover, a full 3D variography (if my 2. comment is valid) may give different insights and results.*

We fully agree on this comment. In fact, this is a very good example for doing exactly that what you've mentioned (Comment #2) in future work, in order to understand and extract REVs on arbitrary scales (meaning here from μm – couple of cm, i.e. the laboratory scale). As stated on your previous comment, we made quite some efforts to be sure that all samples have been derived and measured in alignment with their intrinsic foliation and textural features. For our analysis, we used the largest possible and available 3D image. By nature of the computed tomography, the "size" of the image is limited by the so-called "field of view", which is a result of the sample positioning relative to the X-ray source and the detector. The field of view is always "linked" to the achieved image resolution. Hence, changing the field of view (i.e. making it larger) would lead to a different (i.e. coarser) resolution. We will discuss this point within the manuscript as clarified within our response to the comment #2.

7. *No histogram of apparent image porosity is displayed, neither from the slices used for the variographic REV nor from the subsets of figure B1, although from that figure we get an idea of the "density" (however there is sampling involved here, I assume). Is it possible that the "cube" porosity - at a given cube size - is also lognormally distributed? Possibly then it could be worth to perform the variographic analysis on a log porosity.*

We will add and discuss the 2D "porosity histogram" as recommended in the given context. Since we discussed the 3D variography before (answers to your comments #2 and #6), we will not re-calculate the variograms with log-Phi data at this point.

*8. For sample S3 the REV is identified at 350 voxels, though only one permeability simulation is conducted. It would be nice to demonstrate that the calculated permeability is somewhat "continuous" by repeating the flow simulations on different subsets of that size of the original microCT image.*

We agree that performing the repeated simulations on different subsets of that REV size for S3 to demonstrate the "continuity" of the permeability estimation is important. However, we do not have the (cost-intensive) Materialize (Belgium) meshing license anymore in order to transform the μ-CT images into COMSOL readable meshes that was used in this project. In order to avoid problems related to software usage, we would like to propose using the currently available toolbox GEODICT, which features a "built-in" mesh generator, for this task only. The basic numerical Navier-Stokes algorithm and the according boundary conditions will be exact the same as documented within the manuscript. Hence there should not be numerical differences (inferred from our previous experience) that would create problems or require an extensive discussion. We will first compare the results with those conducted on the current subsample of S3 with Comsol and then perform the additional simulations on the different subsets of S3 with GEODICT. We could easily do this for the S3 sample only as suggested by the reviewer, to avoid a massive re-modelling of all the derived data with GEODICT. The results will be placed within the discussion section. We would like to discuss, if this is an appropriate way for Reviewer #1, and also Reviewer #2, as well as for the topical Editor.

---

## Author Comment (AC2) · 27 Aug 2020

**Response to the Reviewer #2:**

We thank the reviewer for handling our manuscript. Response to the specific comments is presented below:

***General Comments***
*This paper has made commendable efforts to using a multi-scale, multi-methodological approach for the petrophysical characterization of reservoir sandstones. The strength of the study lies in its multiple datasets generated and used. However, the paper requires improvements before it can be ready for publication.*

***1**.Comment:*
*The main aims/objectives of this study should be made very clear from the start. Are you proposing **multi-methodological approach** for the petrophysical characterization of reservoir sandstones as the best or only **method**? Or what exactly are you aiming for?*

Response:
The objective of the paper is formulated in lines 68-69 in the introduction:
"The present paper provides a detailed description and evaluation of a **multi-methodological** petrophysical approach for the comprehensive multiscale characterization of reservoir sandstones."
The word "**method**" questioned by the reviewer appears in lines 62 and 73 in the introduction in the following context:
Line 62: "Over the past few decades, pore-scale imaging and flow simulations (citations…) have started to serve as a reliable **method** for rock characterization."
Line 73: "The suggested computational workflow enables the identification of Darcy-scale permeability links to an extensive set of geometrical, textural and topological rock descriptors, quantified at the pore scale by deterministic and probabilistic (statistical) **method**s."
These **methods** are the parts of **the multi-methodological approach**, which is specified in lines 69-71 in the Introduction: "The proposed **approach** includes petrography, gas porosimetry and permeametry, mercury intrusion porosimetry, 3D imaging and several kinds of pore-scale modelling.".

***2**.Comment:*
*How the achievement of these aims/objectives contribute to the current knowledge gaps should be clearly discussed in the relevant section of the paper.*

Response:
This contribution of the objectives questioned above is presented in detail in the last paragraph of the introduction (lines 76-82):
"The approach presented herein is especially important for the detection of anisotropy and the identification of its origin at various rock scales. The multi-methodological validation procedure is significant for properly upscaling permeability from the micro scale to the macro scale (Ringrose and Bentley, 2015). This validation, thereby, allows an accurate petrophysical analysis of reservoir sandstones with broad ranges of textural and topological

characteristics. The findings contribute also to the current geological knowledge regarding non-marine sandstones of Lower Cretaceous age (e.g., Akinlotan, 2017; Li et al., 2016; Ferreira et al., 2016) and specifically regarding the studied stratigraphic unit."
Some aspects contributing to the current knowledge on anisotropy and on its impact on the clastic formations, will be extended and several more references in this and other context, will be added in the revised version.

**3**.*Comment:*
*The methods need to be clearly discussed.*

Response: Pls see our response to the comments #11 and #13 below.

**4**.*Comment:* *Many figures require attention.*

Response: The quality of all figures will be improved in the revised version of the paper.

**5**.*Comment*
*I do feel that testing all the proposed conclusions made from this study with sandstones from other places will make these conclusions stronger. If it is possible to have sandstones from other places to test your conclusions, this will be very good. However, if the aims/objectives of the study do not require/permit this, then no problem.*

Response:
We do not have the "Darcy-scale permeability links to an extensive set of geometrical, textural and topological rock descriptors, quantified at the pore scale" (from our objective) derived for other sandstones, to perform a valid comparison. So that the general comparison of one sandstone to another one will just move our paper to the category of the regional studies, which should be avoided. Besides, sandstones feature a big complexity and variability, which seems to be impossible to address properly in this paper.

**Specific Comments**
**6**.*Comment:*
*Title*
*Why is there 'benchmark' in the title? Depending on the revised aims/objectives of the study, the title may require revision.*

Response:
The title could be changed to "Validation of a multi-scale, multi-methodological approach for the petrophysical characterization of reservoir sandstones".
However, because benchmarking is comparing results or processes with the "reference" data or processes, and this is exactly what we perform in the paper for the upscaling task, the change of the title would finally depend on the on our implementation of the comment #8 of Reviewer #1. Eventually, if we will add the results from the additional permeability simulations on other REV size geometries of sample S3, then this would underline the benchmarking of the study even better than before. In this case the title will stay as previously.

***7.****Comment:*
*Introduction*
*An extended literature review is required. This may be part of the introduction or may be a separate section. This is important to discuss the state of the subject matter and to present a framework and context for which current study fits into. The current introduction is short while the aim of this study does not seem to address some of the issues raised (lines 66-67) in previous studies.*

Response: We will extend the introduction, keeping the framework of validation of the multi-methodological study and upscaling.

***8.****Comment:*
*Geological Setting*
*Abbreviation in line 98 needs to be written in full at first time.*

Response: Fe-ox will be changed to "Fe oxide (Fe-ox)" at the first occurrence

***9.****Comment:*
*Fig. 1 needs to be increased in size to make it legible. The quality of 1d needs to be improved. 1a needs lines of latitude and longitude.*

Response: These changes will be implemented

***10.****Comment:*
*Appendix A. It is a bit strange that important geological information is put in an appendix. The key geological information in the appendix needs to be summarized and placed under geological setting. The information presented in this section is too shallow and only focused on a formation. Every relevant geological information about the whole basin and other formations should be included here to give a very good context for the current study.*

Response: Moving the geological information to the appendix was requested by the former editor for refocusing this manuscript to its current scope. Information currently provided by Appendix A will be summarised and placed under the geological settings in the revised version. However, the scope of the geological information will not be extended to other formations, to agree with the aim and the scope of the current refocused manuscript, which does not present the regional study (as it was previously) but rather a validation of the multi-methodological approach.

***11.****Comment:*
*Methods*
*3.1. How many samples were collected? Is it possible to state the size of these samples and large block samples or show their photos so that readers can have an idea of how big/small they are. There needs to be proper descriptions of all these samples: how can*

*a reader identify/differentiate a sample from large block samples and from a sub-sample?*

Response: The reviewer is invited to look at lines 119-124 at the manuscript where the information about the number of samples is presented:

"Large sample blocks were collected from these three layers, and the directions perpendicular to the bedding planes (defined as the z-directions in our study) were noted. Subsequently, in the laboratory, smaller sub-samples (**described below**) were prepared from these large samples for textural observations and various analytical measurements and computations. In total, 7 sub-samples from the top layer, 8 sub-samples from the middle layer and 4 sub-samples from the bottom layer were investigated in the laboratory (Table 2)." The information about the number of samples for each test is also indicated in Table 2 and also below Table 2 in the legend.

With respect to the sample sizes: The reviewer is correct, the approximate size of the largest blocks (10÷20 cm) retrieved from the rock at the outcrop is not specified in the manuscript, it will be inserted in the revised version. However, all sample sizes and their shapes used for the specific measurements are specified in the manuscript:

Lines 129-131: "Specimens ~5-7 cm in size were investigated by petrographic and petrophysical lab methods. Sub-samples ~1 cm in size were retrieved from the aforementioned plugs for investigation by 3D imaging, digital image analysis and simulation techniques (described in more detail below)."

The sizes were repeated further in the manuscript at the descriptions of the specific measurements:

Lines 142-143: "Effective porosity and permeability were evaluated on dried cylindrical samples (2.5 cm in diameter and 5-7 cm in length)"

Lines 147-148: "Mercury intrusion porosimetry (...) was applied to dried cylindrical samples ~1 $cm^3$ in size"

Lines 169-170: "cylindrical subsamples 4-8 mm in diameter and 5-10 mm in length were retrieved from the larger samples studied in the laboratory and were scanned..."

With respect to the photos suggested by the Reviewer: because of the big difference in the samples sizes (specified above) and because their dimensions are clearly and repeatedly specified in the paper, we will not insert their photos into the revised version of the manuscript.

*12.Comment:*
*Table 1. 3.7 should be 'Optical microscopy'*

Response: Table 1, point 7 "Petrographic microscopy" will be changed to "Optical microscopy"

**13**.*Comment:*
*3.2 The laboratory methods are not properly discussed and this is not good enough. More than just mentioning the names of equipment used, the procedure needs to be properly discussed or appropriate references provided.*

Response: Methods 1-7 specified in Table 1 are the "classical" ones with well-established protocols available elsewhere. We will add a brief introduction to the Methods section and specify more references, in order to point this out more precisely.

> **_14_**._Comment:_
> _If the methods are properly discussed, I do not see any need for Table 1. Only the_
> _relevant information needed to understand the procedure for the workflow method_
> _should be provided. The current format appears to be excessive._

Response: Extended computational workflow (number 8 in Table 1, Fig.2) is one of the main methodologies of our study. It combines a number of methods with some variability in their application which is not obvious (e.g., especially with respect to the filtering, segmentation, and REV estimation). Despite this, some of these methods (Fig.2a-2c) are described in the text in very brief, e.g. see lines 165-185. REV estimation demands an especial attention in the current paper due to its importance for the anisotropy estimation (see lines 191-215 and comment #2 of Reviewer 1). Flow modelling could also be applied in several ways, with respect e.g. to the boundary conditions and to the averaging procedures. However, those are currently described in brief as well (lines 216-237). Some text from the image analysis (specified in lines 238-257) will be moved into the appendix in the revised version, following the reviewer suggestion.
Table 1 summarizes methods and petrophysical characteristics determined from the these methods (similarly to Table 1 in Tatomir et al. (2016) focusing on the similar rock). This allows an immediate comparison between the output of the corresponding methods. This will be clarified in the text and in the legend to the Table 1.

> **_15_**._Comment:_
> _Results Line 269, 314, -what heavy minerals?_

Response: "heavy minerals" will be replaced with "Fe/Fe-ox bearing minerals"

> **_16_**._Comment:_
> _268, 270: referencing methods using 'according to', 'following' should be amended_
> using journal style

Response: This paper was edited by the professional AJE editorial agency (certificate # 13B3-B361-ED59-44F5-4FB0, attached to this response) in accordance with SE journal style.

> **_17_**._Comment:_
> _276; Mn-Ox: what is this? Please explain?_

Response: Mn-ox is the manganese oxide, which will be clarified in the text in the same way as for Fe-ox before (your comment #8).

> **_18_**._Comment:_
> _317-include reference_

Response: An appropriate reference with a classification of the "quartz wacke sandstone" (Pettijohn et al., 1987) will be included.

*19.Comment:*
*318: result is mixed with interpretation. Only the results should be presented in the result section in this place and throughout the manuscript.*

Response: The sentence "The pore network is influenced by the extent of clay deposition on coarser grains, identified mostly in laminae (Fig. 4a, d)." will be substituted by "The pore space is reduced by clays deposited on coarser grains, identified mostly in laminae (Fig. 4a, d)"

*20.Comment:*
*Fig 4d-scale is missing*

Response: The scale will be added in the revised version

*21.Comment:*
*347-349 should be moved to the methods section*

Response:
To agree with the corresponding descriptions of the top and intermediate unit layers in the Results section:
"**Sandstone S1**: The top unit layer with a thickness of ~1.5 m (Fig. 1c) consists of yellow-brown sandstone (Fig. 3a), which is moderately consolidated ..." (lines 267-268)
"**Sandstone S2**: The intermediate unit layer with a thickness of ~20 cm consists of grey-green moderately consolidated sandstone (Figs. 1c, 4) ..." (lines 310-311),

the following sentence for the bottom unit layer, addressed by the reviewer:
"**Sandstone S3**: Samples were taken from the ~1.5 m thick bottom unit layer in the outcrop (Fig. 1c) consisting of (pale) red-purple poorly consolidated sandstone with grains covered by a secondary red patina (Fig. 5)." (lines 347-349)
will be changed to:
"The bottom unit layer with a thickness of ~1.5 m consists of (pale) red-purple poorly consolidated sandstone (Fig. 1c) with grains covered by a secondary red patina (Fig. 5)."

*22.Comment:*
*Fig 6 and 7, 9, 11, 12 should be increased*

Response: The quality of these figures will be improved in the revised version of the manuscript

*23.Comment:*
*480-485: more or less a repetition. Any new information here should be moved to methods section. The results of the modelling should be presented here.*

Response: The questioned text from lines 480-483 is presented below:

"Fluid flow was modelled at the pore scale in two different micro-CT-scanned geometries: 1) a full cube of sample S1, including two adjacent parts possessing relatively low (0-250 voxels) and high (250-1180 voxels) porosities (Fig. 9c), and 2) sample S3 within its REV dimensions (Table 3). Modelling of the 3D geometry of sample S2 was not performed due to the reasons detailed above."

This text from the first paragraph of the subsection 4.3 on Flow modelling can not be moved to the Methods section as it relies on the results of the REV analysis presented in the preceding subsection 4.2 of the Results. The following sentence "A constant pressure gradient of 2.424 [$Pa / mm$] between the inlet and outlet boundaries was applied in all the simulations for consistency." will be moved to the Methods to the description of the flow modelling.

*24.Comment:*
*509-511: needs to be in the methods section*

Response: The questioned text from lines 509-511 is presented below: "For S1, the mode peak of the pore size distribution (measured by a Feret maximum calliper) (Fig. 13, red line) is at 194 µm (Table 2). In total, 3500 pores were analysed. The pore specific surface area (*PSA*) calculated from micro-CT images is 0.068 µm$^{-1}$."

The knowledge presented in these sentences, including a number of the analyzed pores, is the direct result of the application of the image analysis (see lines 239-241 in Methods section). These are not known before running the model of the image analysis. Hence, they should stay in the Results section.

*25.Comment:*
*513-514: is this result or interpretation?*

Response:

"The tortuosity, measured from the whole CT image, indicates similar values in the x- and y-directions of 1.37 and 1.38, respectively, whereas in the z-direction, the tortuosity is 1.48 (Table 2). As many paths were considered, we suggest that this difference is created by the textural features that appear in horizontal planes (Fig. 3a)."

Both sentences include the results of the conducted image analysis indicating an anisotropy. We will change the second sentence to "As many paths were considered, this difference is an indication of the textural features that appear in horizontal planes (Fig. 3a)."

*26.Comment:*
*545: if the information in appendix C is important for the discussion, why is it not*
*included in the main body of the manuscript?*

Response:
Appendix C presents the definition of the Euler characteristic available elsewhere and used in the image analysis. It was excluded from the main text in order to reduce the amount of

text related to the image analysis in the Methods section (lines 238-258) (see our response to reviewer's comment #14 above). Because these two requests contradict each other, we decided to leave this text in the Appendix C and thus to shorten the Methods section.  In addition, there is also no need to insert this "basic" definition to the Discussion. Therefore, it will be just referenced, as below.

"The value of the connectivity index of S3 (10) is approximately three times higher than that of S1 (3.49), while both rocks are defined as moderately sorted sandstones (Table 2). This difference is due to S1 having a smaller number of inequivalent loops within the pore network than S3 (**Appendix C**), leading to smaller β1 values in Euler characteristics"

*27.Comment:*
*560: gravity-dominated?, capillary-dominated?*

Response: SE English guidelines do not allow using hyphens in the specified grammar context: https://www.solid-earth.net/for_authors/manuscript_preparation.html

*28.Comment:*
*601: use 'study' instead of paper, here and throughout the manuscript*

Response: Will be changed throughout the text where applicable

*29.Comment:*
*603: 'very heterogeneous in nature'?*

Response: Will be changed

*30.Comment:*
*References*
*I have not bothered to check the references at this stage. The author needs to ensure that all cited references are in the bibliography and vice. For example, I am not sure if I encountered Akinlotan 2018 in the text but it is in the bibliography. Please look into this and others and ensure referencing is accurate.*

Response: The list of the reference will be verified and adjusted accordingly in the revised version of the manuscript.

**References**

Pettijohn F. J., P.E. Potter and R. Siever, 1987, *Sand and sandstone*, 2nd ed. Springer-Verlag. ISBN 0-387-96350-2.

---

## Author Response (AR1)

**Referee #1: Referee's comments are in blue, authors' response is in black**

We thank the referee for his comments that helped to significantly sharpen our manuscript. We fully implemented the suggested changes.

The following main changes were incorporated: Description of the semivariogram analysis was incorporated in the Methods Sect. (see below); REV analysis in the Results Sect. was extended by the semivariogram analysis, including the new figures (see comments 2, 6-8 below); the evaluation of the permeability tensors at the multiple sub-volumes of S3 and on the entire sample (comment 8) are discussed in the Discussion Sect. and included in Appenix C; Conclusions and Abstract are adjusted accordingly (see the details below),

1. *Please use "semivariogram" instead of "variogram", or state in the text that you are using "variogram" to intend "semivariogram", as I will do from now on in this review.*

> The "variogram" was changed to "semivariogram" throughout the entire manuscript as suggested.

2. *Is the support used to compute the experimental variograms the whole image plane orthogonal to the computed direction? It is not completely clear to me at the moment.*
   *I find the Methods description misleading in this regard. If it's the case, I strongly advice to use for variography at least squares and not the full slices, and even better yet would be to do a full 3D variography, doing cubic supports as you did in the "classical" REV analysis of figure B1. Though, this is much more computationally intensive and may require ad-hoc coding. This may have a potentially large effect on the results, depending on the chosen "box side", especially for the sample S2, and if done properly also actually hint at the true principal axes of anisotropy for the samples, which may not be aligned with their sides.*

> Methods Sect. was extended by description of the conducted semivariogram analysis (see lines 326-495). REV analysis in the Results Sect. was extended by the semivariogram analysis (lines 871-1788).

> Sample sizes, at which the 1D porosity profiles in each x- y and z- direction are calculated and the semivariogram analysis is conducted, as indicated in lines 873, 1157, 1299 (Sect. 4.3.1), for S1, S3, and S2, respectively.

> In addition, following your suggestion, we used multiple support sub-volumes of different sizes for S2, and evaluated them for the stationarity, variance and for the range of correlation (see lines 1571-1695 and Fig.19). Figure 9 was added for a 3D visualization of the CT segmented images of the samples.

> As it was indicated previously, we did not calculate the semivatiogram for 1D porosities in directions different from the main orthogonal axes of the cube. We made large efforts to take the samples in the field as accurate as possible to achieve a sub-sampling procedure in line with the visible foliation of the different layers (see lines 387-389). The evidence that the resulting "coordinate system" of the samples and of their images are in a very good alignment with the true principle axis of anisotropy, is presented by the exceptionally good results of measured and modelled permeability values and tensors (Table 2).

> Accordingly, we have chosen the "less challenging" 2D variography path for our study, since we are familiar with all the features and textures of the rock samples' very well. The histogram variances and ranges are larger for z-direction (e.g. Figs.11-14, 17-18). Moreover, for the smaller sub-volumes, non-stationarity is also more apparent in z direction (Fig. 19). This confirms our initial assumption of the principal direction perpendicular to the deposition plane.

> We believe that the application and comparison of results from the different variography methods is a big stand-alone topic. It will be of a great impact and thus deserves to be addressed within a new topical manuscript applicable to samples with various (not known) directions of the principal anisotropy axes.

3. *Fig 9-11: if you are showing the fitted variograms, based upon which you define the apparent ranges, I believe you should also state which variogram model was fitted. From the legend I may assume it's an exponential model but or spherical or something else, and the "expo. fit". If it's the case, then you need to specify if the reported "range" is the actual coefficient in the exponential model or the "practical range" of the asymptotic function.*

Now the (nested) variogram models (see description in lines 377-386) and their corresponding calibrated sills and ranges are reported on the figure panels (e.g., Figs. 12, 14, 18).

4. *All three samples represents exemplary cases for zonal anisotropy, where the sills of the variograms are not constant following different directions. This reinforce my suggestion of making the samples available to the public.*

All data are available at the PANGÄA-repository as explained in the "Supplementary materials Sect. S1". The doi is also added at the end of the manuscript, lines 2302-2304.

5. *It is to me however striking - and this may hint to a too large support definition, cfr comment 2, or else to a graphical imprecision - that no experimental variogram displays any nugget effect. This could mean that the variable has been excessively regularised. Please state in the text how the lags for the calculation of the experimental variagrams were chosen, and if the computed pairs at each lag bin are comparable.*

Obviously, the nugget effect can be attributed to measurement errors or spatial sources of variation at distances smaller than the sampling interval or both. Measurement error occurs because of the error inherent in measuring devices. Natural phenomena can vary spatially over a range of scales. Variations at microscales smaller than the sampling distances will appear as part of the nugget effect. Nevertheless, this is more or less impossible to achieve for "mining" spatial data from μ-CT images since we have a fixed resolution limit. Hence, all variation below that "hard resolution boundary" is "invisible" for the variography analysis. This clearly is a drawback of the image analysis, and hence it is important to gain detailed understanding of the scales of spatial variation from multiple methods. Accordingly, the smallest lags are related to the resolution, i.e. the smallest segmented feature of the 3D scan.
These issues are addressed in lines 338-342, 879, 1154, 1279, 1565.

6. *Fig 11. Regarding the variograms of sample S2, they are clearly linear, especially the xy plane, which is a clear sign of non-stationarity, as also clear from the strong trend in subfigure (c). However, you also correctly recognised the "external drift" represented by the clay content. This is possibly a textbook example of external drift, which makes the de-trending of porosity worth. My point here is that the fact that the sample is clearly strongly anisotropic and non-stationary does not mean that it is not possible to extract a REV from it, at least for the two other directions, but with some manipulation, also in the xy plane. Moreover, a full 3D variography (if my 2. comment is valid) may give different insights and results.*

New semivariogram analysis was conducted (see our response to the comment #2 above). Figures of slice-by-slice porosity 1D profiles were improved (Figs.11,13, 15). The trends in the data were modelled and removed, and the residuals are presented (Figs. 11, 17). The histograms of the 1D profiles are now presented (Figs.11,13, 17). Semivariograms (Figs. 12, 14, 18) were calculated from the standardized histograms.

Wherever hole-effect was identified, it was modelled as a part of a nested semivariogram model (e.g. Figs.12, 18).

7. *No histogram of apparent image porosity is displayed, neither from the slices used for the variographic REV nor from the subsets of figure B1, although from that figure we get an idea of the "density" (however there is sampling involved here, I assume). Is it possible that the "cube" porosity - at a given cube size - is also lognormally distributed? Possibly then it could be worth to perform the variographic analysis on a log porosity.*

The histograms of porosity are now added (see Figs. 11,13, 17). They show a normal distribution. Therefore, there is no need to calculate the semivariograms on log porosity.

8. *For sample S3 the REV is identified at 350 voxels, though only one permeability simulation is conducted. It would be nice to demonstrate that the calculated permeability is somewhat "continuous" by repeating the flow simulations on different subsets of that size of the original microCT image.*

These calculations were performed. The results demonstrating a "continuity" of permeability in the different REV sub-volumes and in the entire segmented volume are presented in Appendix C and addressed in lines 1942-1955, 2115-2120 in the Discussion Sect.
An outlook on relation of the spatial variability of structures (semivariogram range of correlation) at the different scales to the spatial distribution of local permeability connected to generation of preferential flow paths, is added (lines 2185-2200).

**Response to the Referee #2:**

We thank the reviewer for handling our manuscript. Our response to the specific comments is presented below:

***General Comments***
*This paper has made commendable efforts to using a multi-scale, multi-methodological approach for the petrophysical characterization of reservoir sandstones. The strength of the study lies in its multiple datasets generated and used. However, the paper requires improvements before it can be ready for publication.*

>  ***1.Comment:***
>  *The main aims/objectives of this study should be made very clear from the start. Are you proposing **multi-methodological approach** for the petrophysical characterization of reservoir sandstones as the best or only **method**? Or what exactly are you aiming for?*

Response:
The objective of the paper is formulated in lines 106-107 in the introduction:
"The present paper provides a detailed description and evaluation of a **multi-methodological** petrophysical approach for the comprehensive multiscale characterization of reservoir sandstones."
The word "**method**" questioned by the reviewer appears in lines 79 and 111 in the introduction in the following context:
Line 79: "Over the past few decades, pore-scale imaging and flow simulations (citations…) have started to serve as a reliable **method** for rock characterization."
Line 111: "The suggested computational workflow enables the identification of Darcy-scale permeability links to an extensive set of geometrical, textural and topological rock descriptors, quantified at the pore scale by deterministic and probabilistic (statistical) **method**s."
These **methods** are the parts of **the multi-methodological approach**, which is specified in lines 107-109 in the Introduction: "The proposed approach includes petrography, gas porosimetry and permeametry, mercury intrusion porosimetry, 3D imaging and image analysis, semivariogram analysis and flow modelling at the pore-scale."

>  ***2.Comment:***
>  *How the achievement of these aims/objectives contribute to the current knowledge gaps should be clearly discussed in the relevant section of the paper.*

Response:
This contribution of the objectives questioned above is presented in detail in the last paragraph of the introduction (lines 114-120):
"The approach presented herein is especially important for the detection of anisotropy and the identification of its origin at various rock scales. The multi-methodological validation procedure is significant for properly upscaling permeability from the micro scale to the macro scale (Ringrose and Bentley, 2015). This validation, thereby, allows an accurate petrophysical analysis of reservoir sandstones with broad ranges of textural and topological characteristics. The findings contribute also to the current geological knowledge regarding non-marine sandstones of Lower Cretaceous age (e.g., Akinlotan, 2016; 2017; 2018; Li et al., 2016; Ferreira et al., 2016) and specifically regarding the studied stratigraphic unit."

Some aspects contributing to the current knowledge on anisotropy are added to the introduction (lines 87-105), and several more references in this and other context, are added to the revised version (lines 81-82, 86, 119, etc.).

*3.Comment:*
*The methods need to be clearly discussed.*

Response: Pls see our response to the comments #11 and #13 below.

*4.Comment: Many figures require attention.*

Response: The quality of all the figures is improved in the revised version of the paper.

*5.Comment*
*I do feel that testing all the proposed conclusions made from this study with sandstones from other places will make these conclusions stronger. If it is possible to have sandstones from other places to test your conclusions, this will be very good. However, if the aims/objectives of the study do not require/permit this, then no problem.*

Response:
In this paper we gathered an extensive data set quantified on various scales on the studied sandstones to benchmark the approach (from our objective). We do not have these data for other sandstones, to perform a valid benchmarking and comparison. However, we added some brief review on the relevant properties of non-marine sandstones of Lower Cretaceous age from other places to the Discussion Sect. (lines 2047-2062).

**Specific Comments**
*6.Comment:*
*Title*
*Why is there 'benchmark' in the title? Depending on the revised aims/objectives of the study, the title may require revision.*

Response:
Benchmarking is comparing results or processes with the "reference" data or processes. This is exactly what we perform in the paper for the upscaling task. Moreover, following the suggestions of Ref.1, this benchmarking was strengthened by performing a semivariogram analysis (lines 871-1695) and by the additional calculations of permeability on the various REV subvolumes of S3 and on the full sample, to show their continuity (lines 1942-1955, 2115-2120 and Appendix C). This strengthens the "benchmarking" content of the study even better than before and thus should remain in our title.

*7.Comment:*
*Introduction*
*An extended literature review is required. This may be part of the introduction or may be a separate section. This is important to discuss the state of the subject matter and to present a framework and context for which current study fits into. The current introduction is short while the aim of this study does not seem to address some of the issues raised (lines 66-67) in previous studies.*

Response: The introduction is extended by addressing some knowledge on anisotropy in sandstones (lines 87-105), and by adding several more references in this and other context (lines 81-82, 86, 119, etc.).

*8.Comment:*
*Geological Setting*
*Abbreviation in line 98 needs to be written in full at first time.*

Response: Fe-ox is changed to "Fe oxide (Fe-ox)" at the first occurrence (line 178).

*9.Comment:*
*Fig. 1 needs to be increased in size to make it legible. The quality of 1d needs to be improved. 1a needs lines of latitude and longitude.*

Response: These changes are implemented.

*10.Comment:*
*Appendix A. It is a bit strange that important geological information is put in an appendix. The key geological information in the appendix needs to be summarized and placed under geological setting. The information presented in this section is too shallow and only focused on a formation. Every relevant geological information about the whole basin and other formations should be included here to give a very good context for the current study.*

Response: The information that was provided in the former Appendix A is moved to the main text (see lines 141-171), following the request of the reviewer. However, to agree with the aim and the scope of the current manuscript (see our response to comment 6 above), we avoid from adding an extensive geological content to our paper.

**11**.*Comment:*
*Methods*
*3.1. How many samples were collected? Is it possible to state the size of these samples and large block samples or show their photos so that readers can have an idea of how big/small they are. There needs to be proper descriptions of all these samples: how can a reader identify/differentiate a sample from large block samples and from a sub-sample?*

Response: The reviewer is invited to look at lines 200-205 at the manuscript where the information about the number of samples is presented:
"Large sample blocks **of ~10÷20 cm size** were collected from these three layers, and the directions perpendicular to the bedding planes (defined as the z-directions in our study) were noted. Subsequently, in the laboratory, smaller sub-samples (**described below**) were prepared from these large samples for textural observations and various analytical measurements and computations. In total, 7 sub-samples from the top layer, 8 sub-samples from the middle layer and 4 sub-samples from the bottom layer were investigated in the laboratory (Table 2)." The information about the number of samples for each test is also indicated in Table 2 and also below Table 2 in the legend.
With respect to the sample sizes: The approximate size of the largest blocks (10÷20 cm) is added to the manuscript (in bold above, line 200). However, all sample sizes and their shapes used for the specific measurements are specified in the manuscript:
Lines 210-212: "Specimens ~5-7 cm in size were investigated by petrographic and petrophysical lab methods. Sub-samples ~1 cm in size were retrieved from the aforementioned plugs for investigation by 3D imaging, digital image analysis and simulation techniques (described in more detail below)."
These sizes were repeated further in the manuscript at the descriptions of the specific measurements:

Lines 237-238: "Effective porosity and permeability were evaluated on dried cylindrical samples (2.5 cm in diameter and 5-7 cm in length)"
Lines 242-243: "Mercury intrusion porosimetry (...) was applied to dried cylindrical samples ~1 cm$^3$ in size"
Lines 273-274: "cylindrical subsamples 4-8 mm in diameter and 5-10 mm in length were retrieved from the larger samples studied in the laboratory and were scanned...".
Sample sizes for the semivariogram analysis are also specified in the text (see lines 873, 1157, 1299).

With respect to the photos suggested by the Reviewer: because of the big difference in the samples sizes (specified above) and because their dimensions are clearly and repeatedly specified in the paper, we did not insert their photos into the revised version of the manuscript.

*12.Comment:*
*Table 1. 3.7 should be 'Optical microscopy'*

Response: Table 1, point 7 "Petrographic microscopy" is changed to the "Optical microscopy"

**13**.*Comment:*
*3.2 The laboratory methods are not properly discussed and this is not good enough.*
*More than just mentioning the names of equipment used, the procedure needs to be*
*properly discussed or appropriate references provided.*

Response: Methods 1-7 specified in Table 1 are the "classical" ones with well-established protocols available elsewhere. We stated this more clearly in lines 213-217 and cited a basic comprehensive reference of *Practices for Core Analysis, API, (1998)*. We also added the additional references to some of the laboratory methods in lines 228, 235, 239, 243.

*14.Comment:*
*If the methods are properly discussed, I do not see any need for Table 1. Only the relevant information needed to understand the procedure for the workflow method should be provided. The current format appears to be excessive.*

Response: Extended computational workflow (number 8 in Table 1, Fig.2) is one of the main methodologies of our study. It combines several methods with some variability in their application which is not obvious (e.g., especially with respect to the methods of filtering, segmentation, and REV estimation). Despite this, some of these methods (Fig.2a-2c) are described in the text in very brief, e.g. see lines 271-289. REV estimation demands an especial attention in the current paper due to its importance for the anisotropy and inhomogeneity estimation (emphasized by Ref. 1 in his comment #2). Following this request, we added some explanations regarding the semivariorgam analysis in lines 326-495 and also added this method to Table 1. Flow modelling could also be applied in several ways, with respect e.g., to the boundary conditions and to the averaging procedures. However, those are currently described in brief as well (lines 496-543). Image analysis description is very concise (lines 544-550). It contains a necessary information on algorithms and software that allows the reader to repeat the process using the data on our samples (available online, see Supplementary material for more detail).
The necessity of Table 1 is clarified in lines 564-566.

*15.Comment:*

*Results Line 269, 314, -what heavy minerals?*

Response: The clarifications are added in lines 582, 644.

**16**.*Comment:*
*268, 270: referencing methods using 'according to', 'following' should be amended*
using journal style

Response: This paper was edited by the professional AJE editorial agency (certificate # 13B3-B361-ED59-44F5-4FB0, attached to this response) in accordance with SE journal style.

**17**.*Comment:*
*276; Mn-Ox: what is this? Please explain?*

Response: Mn-ox is the manganese oxide, which is clarified in the text in the same way as Fe-ox before (your comment #8), see line 589.

**18**.*Comment:*
*317-include reference*

Response: An appropriate reference with a classification of the "quartz wacke sandstone" (Pettijohn et al., 1987) is included in line 647.

**19**.*Comment:*
*318: result is mixed with interpretation. Only the results should be presented in the result section in this place and throughout the manuscript.*

Response: The sentence "The pore network is influenced by the extent of clay deposition on coarser grains, identified mostly in laminae (Fig. 4a, d)." is substituted by "The pore space is reduced by clays deposited on coarser grains, identified mostly in laminae (Fig. 4a, d)", see lines 648-649.

**20**.*Comment:*
*Fig 4d-scale is missing*

Response: The scale is added

**21**.*Comment:*
*347-349 should be moved to the methods section*

Response:
To agree with the corresponding descriptions of the top and intermediate unit layers in the Results section:
"**Sandstone S1**: The top unit layer with a thickness of ~1.5 m (Fig. 1c) consists of yellow-brown sandstone (Fig. 3a), which is moderately consolidated ..." (lines 580-581)
"**Sandstone S2**: The intermediate unit layer with a thickness of ~20 cm consists of grey-green moderately consolidated sandstone (Figs. 1c, 4) ..." (lines 640-641),
the following sentence for the bottom unit layer, addressed by the referee:
"**Sandstone S3**: Samples were taken from the ~1.5 m thick bottom unit layer in the outcrop (Fig. 1c) consisting of (pale) red-purple poorly consolidated sandstone with grains covered by a secondary red patina (Fig. 5)." (former lines 347-349)

are changed to:

"The bottom unit layer with a thickness of ~1.5 m consists of (pale) red-purple poorly consolidated sandstone (Fig. 1c) with grains covered by a secondary red patina (Fig. 5).", see lines 693-694.

*22.Comment:*
*Fig 6 and 7, 9, 11, 12 should be increased*

Response: The quality of all the figures in the manuscript is improved

*23.Comment:*
*480-485: more or less a repetition. Any new information here should be moved to*
*methods section. The results of the modelling should be presented here.*

Response: The questioned text from the lines 480-483 in the former version of our ms is presented below:
"Fluid flow was modelled at the pore scale in two different micro-CT-scanned geometries: 1) a full cube of sample S1, including two adjacent parts possessing relatively low (0-250 voxels) and high (250-1180 voxels) porosities (Fig. 9c), and 2) sample S3 within its REV dimensions (Table 3). Modelling of the 3D geometry of sample S2 was not performed due to the reasons detailed above."
The new text in presented in lines 1797-1800. Please note that the domain size for the flow modelling relies on the results of the conducted REV analyses (including the new semivariorgam analysis suggested by Ref.1).
The following sentence "A constant pressure gradient of 2.424 [$Pa / mm$] between the inlet and outlet boundaries was applied in all the simulations for consistency." is moved to the Methods to the description of the flow modelling (lines 504-506).

*24.Comment:*
*509-511: needs to be in the methods section*

Response: The questioned text in lines 509-511 of the former version, is presented below: "For S1, the mode peak of the pore size distribution (measured by a Feret maximum calliper) (Fig. 13, red line) is at 194 μm (Table 2). In total, 3500 pores were analysed. The pore specific surface area (*PSA*) calculated from micro-CT images is 0.068 μm$^{-1}$."
The number of the analyzed pores is the direct result of application of the image analysis (see lines 544-550 in Methods section). Now the number of the derived pores is specified in the caption to Fig.10. All other parameters derived by image analysis are presented in lines 784-788.

*25.Comment:*
*513-514: is this result or interpretation?*

Response:
"The tortuosity, measured from the whole CT image, indicates similar values in the x- and y-directions of 1.37 and 1.38, respectively, whereas in the z-direction, the tortuosity is 1.48 (Table 2). As many paths were considered, we suggest that this difference is created by the textural features that appear in horizontal planes (Fig. 3a)."
Both sentences include the results of the conducted image analysis indicating an anisotropy. The second sentence is changed to "As many paths were considered, this difference is an indication of the textural features that appear in horizontal planes (Fig. 3a).", see lines 789-791.

***26***.*Comment:*
*545: if the information in appendix C is important for the discussion, why is it not*
*included in the main body of the manuscript?*

Response:
Former Appendix C presented the definition of the Euler characteristic available elsewhere and used in our image analysis, with some clarification. It was excluded from the Methods section in order to reduce the text related to the image analysis (see comment #14 above and our response). In addition, because there is also no need to insert this "basic" definition to the Discussion, we decided to place this brief text to the Supplementary material and to refer to it from the main text (see lines 558, 1856).

***27***.*Comment:*
*560: gravity-dominated?, capillary-dominated?*

Response: SE English guidelines do not allow using hyphens in the specified grammar context:
https://www.solid-earth.net/for_authors/manuscript_preparation.html

***28***.*Comment:*
*601: use 'study' instead of paper, here and throughout the manuscript*

Response: This was changed throughout the text where applicable

***29***.*Comment:*
*603: 'very heterogeneous in nature'?*

Response: Changed, see line 2039.

***30***.*Comment:*

defines the relation between a spatially varying property (porosity in our case) and a lag distance (a Cartesian
distance between the points in the studied domain). The semivariogram value increases when the values of the
studied parameter become more dissimilar. Semivariogram, $\gamma(h)$, is based on the difference in values of the
studied property between all combinations of pairs of data points, $x_i$ and $y_i$, in the studied domain and is defined
by:

$$\gamma(h) = \frac{1}{2N(h)}\sum_{i=1}^{N(h)}(x_i - y_i)^2 \hspace{3cm} (2),$$

where $N(h)$ is the number of pairs, and $h$ is the lag distance. A semivariogram describing a single variability
structure is characterized by a sill (a plateau) in which $\gamma$ corresponds to the total variability of the sample, and
by the range (of correlation) in $h$ at which $\gamma$ reaches the sill value (usually ~95 % of the sill value). Due to the
spatial continuity of the CT dataset, a lag increment of a voxel size set by the CT acquisition resolution, was
chosen for calculation of the semivariogram, thus producing a large amount of data point pairs at each lag to
get a significant mean (Ploner, 1999). A non-zero intercept in the semivariogram (nugget effect) may exist
due to variability at lengths smaller than the lag distance and due to noisy data. For CT data, natural variations
below the fixed resolution limit are invisible for the variography analysis. Therefore, in CT data the nugget
effect may be observed only due to measurement error inherent to measuring devices. In a case they are small,
the semivariogram will approximately have a zero intercept.

There are several analytical models that fit the semivariograms. In this study we use three of them (for more information see Cressie, 1985).

Gaussian: $\gamma(h) = C\left[1 - \exp\left(-3\left(\frac{h}{a}\right)^2\right)\right]$     (3)

Spherical: $\gamma(h) = C\left[\frac{3}{2}\frac{h}{a} - \frac{1}{2}\left(\frac{h}{a}\right)^3\right]$ for $h<a$, $C$ else.     (4)

Hole-effect: $\gamma(h) = C\left[1 - cos\left(\frac{h}{a}\pi\right)\right]$     (5)

where $C$ is the calibrated sill and $a$ is a calibrated parameter.

Multiple sills may be associated with different variability structures, which are characteristic, for instance, for the different scales. Nested sills are modelled as a linear combination of the single models:

Nested sills: $\gamma(h) = \sum_{i=1}^{n} C_i \gamma_i(h)$, $0<C$     (6)

where $C_i$ is the contribution of a single sill.

z-direction of CT specimen used in this analysis is perpendicular to the natural layering of the sandstone identified in the outcrop and in the petrographic observations. x- and y- orthogonal directions lie in the horizontal plane, with an azimuth chosen randomly. The application of the semivariogram analysis using all data points (the voxels) distributed at multiple 3D sub-volume domains, is computationally intensive as the typical CT dataset includes $10^9$ points and more. To allow faster semivariogram calculations, we slice the volume by cross-sections with a voxel size distance between them in each direction and evaluate the porosity at each cross-section, which produces a one-dimensional porosity profile. This results in histograms (the population of cross-sections porosity) that differ in each direction and also from those in the original 3D

dataset. Therefore, for each direction a semivariogram model is independently calculated and modelled. In case when the variable is identified with a spatial systematic trend, the mean value will not to be independent of location (considered as a non-stationarity) that disconsiders the representativeness of the sample. When such a trend is identified, it should be modelled and removed from the property dataset, to remain with residuals.

Next, a common practice is to calculate the semivariogram to a standardized histogram of the data to have a sill of 1 (Gringarten and Deutsch, 2001), which is achieved by z-score transformation (a normal score transform) of the residual porosity histograms. For evaluation of representative lengths, when along some direction the variability changes merely with an increasing lag distance, a sill (in a nested model) is calibrated

[revised manuscript text omitted]

orthogonal directions of S1, having a maximal available segmented volume of *6.8 × 6.9 × 9.2 mm³* scanned
with a voxel size of *5 µm* (suitable for imaging pore throats that effectively contribute to the flow in S1, Table
2). The slice-by-slice porosity distinguishes the z-direction as having an exceptional behaviour, with variance
in porosity fluctuations being four times larger than that in x- and y- directions. Porosity fluctuates around the
mean value with a wavelengths of ~0.2 mm in x- and y- directions. Semivariograms (obtained after z-score
transformation of porosity histograms, Figure 11d) show stationarity in x- and y- directions (i.e., sill~1, Figure
12). The nugget is set to zero (see Methods Sect. 3.2). A semivariogram was fitted using Gaussian model with
range of ~0.1 mm for x- and y- directions (Figure 12a,b). A second semivariogram feature is the waviness,
i.e. cyclic behaviour with lag distance. The lag distances of the first peaks in the experimental variograms are
0.135 and 0.15 mm in x- and y- directions, respectively. In laminated geologic settings, the distance to the
first peak is an indication of the average thickness of the bedding in that direction (Pyrcz and Deutsch, 2003).
However, here this distance refers to the average size of the cross sections of pores in this direction (a cross-
section varies from zero to the maximal one), which is smaller than the average pore size measured in image
analysis. The lag distance between the peaks is about 0.2 mm, being similar to that observed in Figure 11a,b
that refers to the average cross-section of pores and grains in that direction. In z-direction (Figure 11c, black
line) the variability is larger than in x- and y- directions, depicted by a cyclic structure with ~3.5 mm
wavelength. A cyclic structure with lower amplitude and smaller wavelengths in z-direction "rides" on the
main structure with a higher wavelength. A trend in porosity increase with distance is observed, being
modelled by a linear regression model (0.166 % porosity per mm), which was removed from the original porosity, (in black) to obtain the porosity residuals (in red, Figure 11c). The experimental semivariogram of the residuals (Figure 12c, in red) shows smaller sill and range than the semivariogram of the original porosity, (in black). The semivariogram in z-direction was fitted with two nested models, a Gaussian model with range of 0.1 mm similar to those applied to x- and y- directions, and a hole-effect model with range of 1.63 mm. The hole-effect range refers to the average thickness of horizontal layering structures of mm-scale, comprising larger or smaller grains, which impose larger or smaller pores between them (see sample cross-section in microscopy in Figure 3b). The first trough at 3.4 mm lag distance (Figure 112c) in the curve with a larger wavelength, refers to the average thickness of these two layers with higher and lower porosity. A secondary structure in the semivariogram of the lower wavelength of ~0.2 mm is similar to those observed in x- and y-directions. The larger-scale layering is discerned in z-direction only, and the larger variance of porosity in that direction, implies an anisotropy in a sandstone S1.

In order to capture this layering pattern, a volume with a side length of at least ~3.5 mm in z-direction is required. The ranges in x- and y- directions of ~0.1 mm are associated with the typical pore sizes, which are too small to be used in flow modelling, as to predict the permeability reliably, it is necessary to capture the three-dimensional tortuosity of the pore space. Alternatively, the REV size from the classic approach, was estimated as ~1.2 mm (Appendix A, Figures A1a, b). Therefore, for the flow simulations, we decided to use a segmented specimen cube with a maximal available edge length of 2950 μm, scanned with a higher resolution of a 2.5 μm, to preserve a consistency between the flow simulations in S1 and S3.

[Figure]

**_Figure 11:_** _One-dimensional porosity profile of S1 slices evaluated in a) x-direction, b) y-direction and c) z-_

_direction. Investigated volume size is 6.8 × 6.9 × 9.2 mm³. For z-direction, the right y-axis refers to the_

_residual porosity after removing the trend, modelled by a linear regression model. The porosity variations_

_both before (in black) and after (in red) the trend removal are demonstrated. (d) Porosity histograms (z-_

_direction after the trend removal)._

[Figure]

***Figure 12:** Semivariograms of S1 in a) x-direction, b) y-direction and c) z-direction. z-score transform was applied to the histograms (Figure 11d). For z-direction it was applied before (in black) and after (in red) the trend removal (Figure 11c). The variogram analytical models (Gaussian and hole-effect) are also displayed (dashed blue line), modelled with nugget set to zero.*

One-dimensional profiles of porosity were evaluated in sequential slices in the orthogonal directions of S3 with a maximal segmented volume of $3 \times 3 \times 4.2\ mm^3$ scanned with a voxel size of 2.5 $\mu m$ (Figure 13a-c). Porosity fluctuates around the mean with no trend in all directions. Modelling of the experimental semivariograms (Figure 14) shows stationarity in the investigated domain with range of ~0.087 mm, smaller than that in sample S1 (~0.1 mm). Cyclic structure contributes to ~40 % in the semivariogram variability. The first peak in x- and y-directions are at ~0.115 mm and in z-direction it is at ~0.103 mm, which relates to the average size of pore cross-section in that direction. The first trough at ~0.2 mm in all directions, relates to the average cross section of pore and grain together in those directions. Porosity variance in z-direction is ~1.08 that is slightly larger than that in other directions (~ 0.9). However a distinct difference in the spatial variability as that in S1 is not observed, which implies that S3 has lower anisotropy characteristics.

[Figure]

**Figure 13:** *One-dimensional porosity profile of S3 slices evaluated in a) x-direction, b) y-direction and c) z-direction. Investigated maximal segmented volume size is $3 \times 3 \times 4.2$ mm³. d) Porosity histograms.*

| Deleted: measured along |
| Deleted: axis |
| Deleted: axis |
| Deleted: axis |
| Deleted: V |
| Deleted: X |
| Deleted: X |

***Figure 14:*** *Semivariograms of S3 in a) x-direction, b) y-direction and c) z-direction. The experimental semivariogram was modelled using Gaussian model (dashed blue line). The semivariogram models are also displayed, modelled with nugget set to zero.*

Classic REV porosity analysis (Appendix A, Figure A1e,f) yields REV size with a cube edge length of 875 μm (350 pixels), which is about ten-fold larger than the ranges observed in the semivariogram analysis (~0.1 mm, which is a typical size of a pore). For the same reasons used in estimations of REV by the classic approach in S1, the classic REV derived volume was used for the flow modelling in S3.

**4.3.2 Quartz wacke sandstone (S2)**

Sample S2 is more heterogeneous than S1 and S3 because of the deposition of clay. The sample is visualized in Figure 9b with quartz grains (yellow), pore volume (black), clay matrix (brown) and heavy minerals (white). The clay matrix is distributed in patches. In Figure 15, the porosity of sequential planes in the orthogonal directions is shown together with clay matrix content. In z-direction a clear trend in porosity is observed, which has a negative correlation with the clay content (Figure 16), whereas in the horizontal (x-y) plane there is no clear correlation. This correlation in z- direction implies that the porosity is controlled by the depositional processes (However, the similar large-scale wavy structures of the clay content in x- and y-directions (Figures 15a,b) may refer to errors originated from the scanning and inversion in the CT acquisition, as x- and y-coordinates are associated with the side boundaries of the cylindrical sample). The trend was removed in all three directions to remain with residual porosity, when the largest trend slope was in z-direction (Figure 17). After the trend removal, the histogram variance in x- direction appears as the largest one, more than twice larger than that in y-direction. Semivariogram analyses were performed (Figure 18) to investigate the spatial variability of the entire available 3D volume (7.9 × 6.8 × 9.2 m³) scanned with 5 μm resolution. Experimental semivariograms were calculated twice: on porosity values, and on porosity residuals (both the porosity and residuals are z- score transformed before calculating the semivariogram). After the trend removal, the range and sill decrease in all directions and especially in z-direction that now reaches the sill. In x-direction the detrended semivariogram keeps increasing above sill of 1 for the increasing lag distance, thus indicating that the trend was not completely removed, and that the domain is not fully stationary. The semivariograms have multiple structures including cyclicity, therefore, three analytical nested models were used in all directions: Gaussian, spherical and hole-effect models. The smallest range for the y- and z- directions (Figure

18) is within 0.062-0.07 mm, which refers to the average size of pore cross-section, in agreement with the pore size distribution from the image analysis (Figure 10), which is smaller than those of the S1 and S3. However, the contribution of the pore size scale to the overall variability (0.2-0.35) is smaller than that in S1 and S3. The ranges of the larger-scale cyclic structure are ~1.1 mm and ~2.1 mm in horizontal and vertical directions, respectively. These length scales relate to an average thickness of the more porous "lenses" originated at the presence of patchy clay deposition. An intermediate range between those from the Gaussian and hole-effect models in y- and z- directions are of ~0.35 mm discerned by a spherical model, may relate to another structure, associated with a distance between pores.

[Figure]

**_Figure 15:_** _Porosity (left) and clay (right) profiles in slices of S2 evaluated in (a) x-direction, (b) y-direction_

_(c) z-direction. Investigated maximal segmented volume size is_ $7.9 \times 6.8 \times 9.2$ m$^3$ _(see text for more detail)._

[Figure]

[Figure]

**_Figure 16:_** _Scatterplots of clay content and porosity in S2 in (a) x-direction, (b) y-direction, (c) z-directrion._

[Figure]

**_Figure 17:_** _One-dimensional porosity profile of S2 slices estimated in a) x-direction, b) y-direction and c) z-_
_direction. The left vertical axes of the panels refer to the real porosity values, while the right axes to the_

*residual porosity after removing the trend, modelled by a linear model, shown in each subplot. The variances before and after the trend removal are indicated. (d) Porosity histograms after the trend removal.*

[Figure]

***Figure 18:*** *Semivariograms of S2 in a) x-direction, b) y-direction and c) a-direction. z-score transformation was applied to the histogram, before (black) and after (red) the trend removal. The experimental semivariogram was also modelled using nested Gaussian, spherical and hole-effect models (dashed blue line) with nugget set to zero. The calibrated sill and range are shown (see text for more detail).*

Alternatively, the large difference between the mean and median porosities is identified by the classic porosity-based REV approach (Appendix A, Figure A1c, d). Together with the porosity trend in z-direction (Figure 15) in the volume under investigation ($7.9 \times 6.8 \times 9.2$ m³), there is no REV in S2. As a result, flow modelling could not be conducted in sample S2.

Additionally, there is an opportunity to investigate a heterogeneity of S2 by analysing semivariograms for the sequential slice porosity for the multiple sub-volumes of the volume investigated above ($7.9 \times 6.8 \times 9.2$ m³ size, Figure 19). For each sub-volume cube (with 3.5, 1.2, and 0.5 mm edge size) the trend in porosity was removed and the histogram was standardized. A threshold was set such that only the stationary semivariograms were analysed to calibrate the range of correlation. This threshold is associated with sills between 0.9 and 1.1.

For $3.5^3$ *mm³* sub-volumes (Figure 19a-c), with an edge size 2-3 times larger than the cyclic range found (1.1 mm in x- and y- directions, 2.1 mm in z- direction, Figure 18), and which is about half the size of the entire domain ($7.9 \times 6.8 \times 9.2$ mm³), the spherical model was used, and the percentages of passing the threshold were 39, 33 and 12 % in x-, y- and z-directions, respectively. The average residual variances were

0.39, 0.34 and 0.59, respectively. Experimental semivariograms were chosen randomly for the observation (Figure 19a-c). The semivariograms vary in all directions (indicated by the variability in slopes till reaching the ranges), when the calibrated ranges averaged to 204±65 µm, 202 ±69 µm and 334 ± 137 µm, for x-, y- and z- directions, respectively. The large cycles of more than 1 mm in the wavelengths are not consistent. Overall, for $3.5^3$ $mm^3$ sub-volume sizes, the ranges in x- and y-directions do not vary much, but z-direction shows a smaller percentage of sub-volumes passing a sill threshold and a larger variance, which points on more heterogeneous and irregular structure of the pore space.

Modelled $1.2^3$ $mm^3$ sub-volumes (Figure 19d-f) are smaller than the cyclic length calibrated for the entire domain (Figure 18). Gaussian model, which fits the smallest range structures in the entire domain, was used. The percentage of the sub-volumes which passed the threshold were 35, 42 and 17 % in x-, y- and z-directions, respectively. The average residual variances were 1.76, 1,59 and 3.27, respectively. The semivariograms initial slopes (Figures 19d-f) vary mainly in z-direction, when the calibrated ranges averaged to 79±18 µm, 72 ±19 µm and 187 ± 120 µm in x-, y- and z- directions, respectivelly.

For the smallest $0.5^3$ $mm^3$ sub-volumes (Figure 19g-i), the percentages which passed the threshold were 27, 30 and 11 % in x-, y- and z-directions, respectively. The average residual variances were 4.6, 4.8 and 12.3. The semivariogram initial slopes (Figure 19g-i) vary more in z-direction, when the calibrated ranges averaged to 43±8.3 µm, 43 ±9 µm and 91 ± 44 µm in x-, y- and z- directions, respectively.

To summarize, small percentages of sub-volumes which passed the stationarity threshold imply that the sub-volumes of those sizes are not representative for S2. For the increasing sub-volume sizes the range increases (Figure 19), because larger sub-volume is composed of smaller sub-volumes with very different structure characteristics, including those that have not passed the threshold. The larger range identified in z-direction is assumed to be related to an irregular structure in that direction, whereas in x- and y- directions the heterogeneity is milder. The larger porosity variance in z-direction implies an anisotropy due to less constrictions to flow in the horizontal plane.

[Figure]

*Figure 19*: *Experimental semivariograms of sub-volume cubes with $3.5^3$ mm$^3$ (top row), $1.2^3$ mm$^3$ (middle tow) and $0.5^3$ mm$^3$ (bottom row) edge sizes, in x-direction (left column), y-direction (middle column) and z-direction (right column), respectively.*

[revised manuscript text omitted]

Fe-ox.

In this study we used the semivariogram range of spatial correlation of porosity as a parameter to determine the REV from CT data (in addition to the classic method suggested by Bear, 2013). The spatial correlation of porosity relates to a distance that fluid travels without being constrained by grains, and therefore to permeability. Calibrating the range of correlation of porosity by modelling the semivariogram for different sub-volume sizes sheds light on the specimen heterogeneity at the different scales. This approach could be applied for a series of CT datasets, to determine the REV from the range of correlations and to compare to the

REV of permeability. Quantifying the spatial variability of structures which occur at different sub-volume sizes, may link to generation of preferential flow paths and to determination of effective porosity (associated with mobile water fraction) that is available for transport.

**6. Conclusions**

This paper presents a detailed description and evaluation of a multi-methodological petrophysical approach for the comprehensive multiscale characterization of reservoir sandstones. The validation was performed on samples from three different consecutive layers of Lower Cretaceous sandstone in northern Israel. The following conclusions can be drawn:

1. The suggested methodology enables the identification of links between Darcy-scale permeability and an extensive set of geometrical, textural and topological rock descriptors, which are quantified at the pore scale by deterministic and statistical methods. Specifically, micro-scale geometrical rock descriptors (grain and pore size distributions, pore throat size, characteristic length, pore throat length of maximal conductance, specific surface area, and connectivity index) and macro-scale petrophysical properties (porosity and tortuosity), along with quantified anisotropy and inhomogeneity, are used to predict the permeability of the studied layers.

2. Laboratory porosity and permeability measurements conducted on centimetre-scale samples show less variability for the quartz arenite (top and bottom) layers and more variability for the quartz wacke (intermediate) layer. The magnitudes of this variability in the samples are correlated with the dimensions of their representative volumes and anisotropy, both of which are evaluated within the micro-CT-imaged 3D pore geometry. This variability is associated with clay and cementation patterns in the layers and is quantified in this study with image and semivariogram analyses.

3. Two different correlation lengths of porosity variations are revealed in the top quartz arenite layer by statistical semivariogram analysis: fluctuations at ~100 μm are due to variability in grain and pore sizes, and those at ~1.6 mm are due to the occurrence of high- and low-porosity horizontal bands occluded by Fe-ox cementation. The latter millimetre-scale variability is found to control the macroscopic rock permeability measured in the laboratory. Bands of lower porosity could be generated by Fe-ox cementation in regions with higher surface areas adjacent to preferential fluid flow paths.

4. More heterogeneous pore structures were revealed in the quartz wacke sandstone of the intermediate layer. This heterogeneity resulted from a combination of several spatial structures, each one with an internal irregularity: the pore size at the scale of ~50 μm, the distance between the pores at the scale of ~350 μm, and the larger-scale more porous "lense" structures originated at the presence of patchy clay deposition at the ~1-2 mm distance.

5. Quartz arenite sandstone of the bottom layer shows stationarity in the investigated domain and lower anisotropy characteristics than that of the top layer, due to less horizontal cement bands. Modelling of the experimental semivariograms indicates a scale of ~0.1 mm in all directions, associated with the average size of pore cross-section.

[revised manuscript text omitted]

¶

**Appendix C: Setup and results of permeability tensor simulations in sub-samples using GeoDict (module FlowDict)**

[Figure]

S3_full
1200x1200x1200
@2.5 µm

S3 Subvolumes:

S3_#1

S3_#2

S3_#3

S3_#4

S3_#5

Each
350x350x350
voxel
@2.5 µm

*Figure C1: Schematic of sub-volumes location in the full segmented sample of S3.*

[Figure]

**S3_total**

| 4247 | 98 | 116 |
|---|---|---|
| 98 | 4820 | 483 |
| 117 | 483 | 4432 |

avg. k: ˷ 4500 mD
@ ϕ: 26.88 %

**S3_#3**

| 5032 | 9 | 10 |
|---|---|---|
| 1 | 4143 | 10 |
| 2 | 9 | 5223 |

avg. k: ˷ 4799 mD
@ ϕ: 27.33 %

**S3_#1**

| 5485 | 119 | 3 |
|---|---|---|
| 130 | 5882 | 2 |
| 5 | 1 | 4504 |

avg. k: ˷ 5290 mD
@ ϕ: 27.47 %

**S3_#4**

| 4601 | 2 | 3 |
|---|---|---|
| 6 | 3799 | 5 |
| 4 | 5 | 3344 |

avg. k: ˷ 3915 mD
@ ϕ: 26.75 %

**S3_#2**

| 4392 | 179 | 68 |
|---|---|---|
| 75 | 3510 | 185 |
| 175 | 185 | 3359 |

avg. k: ˷ 3754 mD
@ ϕ: 25.99 %

**S3_#5**

| 4397 | 9 | 68 |
|---|---|---|
| 10 | 3825 | 7 |
| 71 | 5 | 3858 |

avg. k: ˷ 4027 mD
@ ϕ: 27.53 %

In sub-samples: avg. k: 4381 mD / median k: 4522 mD

*Figure C2: Results of permeability tensor simulations using GeoDict (module FlowDict).*

---

## Author Response (AR2)

Dear Editor and Reviewer,

Please find below our point-by-point response to reviewer's comments. We are still convinced that the 1D directional variography provides an important information about the anisotropy and heterogeneity of the investigated samples (which has not been used by previous studies and which in fact could presents the "novel" component of our approach), and can assist in characterizing the flow-determined REV substantially. We think that the problem was that our former charts did not include all the necessary details that misled the reviewer. Nevertheless, **we decided to drop it completely**, following the reviewer and editor suggestions (below). We plan to address it systematically and in full detail in a new paper.

In the last sentence of his Comment 6 the reviewer indicates: "*Given the flaws of this variographic analysis it's not surprising to me that the "apparent", interpreted range is ten times smaller than the classical REV determination. The authors should consider either doing it properly, possibly involving some experienced geostatistician in the analysis, or **else leave it all together**.*"

The issue also appears in editor's decision: "*However, the reviewer also notes that you should undertake a careful analysis if the geostatistical investigations are indeed necessary. As they stand now, the results and corresponding interpretation are questionable. **Should you choose to drop the variogram analysis**, the reviewer recommends that the new simulations of permeability within the whole of sample S3 be presented and discussed as part of the main text.*".

We present our detailed response to the reviewer comments point-by-point with additional charts and insets. Our first author performed the additional quite extensive coding and calculations of 3D variography to address their comment #5. We address the simulations in S3 sub-volumes and in the entire sample in our response to comment #7. Figure numbers addressed below in our responses to comments #1-6 correspond to the previous version of our manuscript.

Authors response to reviewer's comments:

Reviewer's Comment 1*:*

In the variograms of figure 12(a) and (b) and all of Fig. 14 I can only recognize pure nugget effect combined with high-frequency fluctuations. The computation of the experimental variograms in terms of retained "bins" and "lags" must be probably adjusted (here I mean: increased to supports larger than these high frequencies fluctuations, and not following an incremental pixel by pixel lags. In other words: the original image should probably be "regolarized" in order to apply variographic analysis) in order to smooth them out. Furthermore, it's not clear which are the calculated experimental values near the origin, since they are covered by the model line. In any case, relying on the fitted range of the gaussian models as measure of correlation length for the porosity in these samples is in my opinion not supported by the experimental evidences. First of all, the gaussian model is always a suspect one to the geostatistician, since realizations of the corresponding random function are infinitely often differentiable (they are analytic functions): this is contradictory with their randomness (see Matheron 1972, C-53, p73-74; available at http://cg.ensmp.fr/bibliotheque/public ). Secondly, a practical argument for rejecting this variogram model in the presented cases is the fact that at the origin it is supposed to have near-horizontal behavior (roughly, an S-shaped function). The calculated experimental variograms do not display such behavior, with possible exception of figure 18(b).

Authors' Response 1:

- Experimental semivariogram values near the origin: we now added to the semivariogram of samples S1-S3 (Figures 12_new, 14_new, 18_new, below) insets that focus on the short lag distances (subplots d-f in the Figures 12_new, 14_new, 18_new below). Now, **it can be seen that near the origin a nugget effect is not observed and the experimental semivariograms are characterized by a parabolic behavior (red dots), which implies a continuity (due to the smoothing occurred when averaging over a slice) of the investigated property and with a continuous slope**. We remind that the data investigated in the semivariograms are an averaged porosity along incremental slices (whereas the averaging is applied on binarized voxel value: '1' – pore or '0' – grain) with a single voxel width along (x-,y-,z-) directions (Figure 11,13,17 in the manuscript). The parabolic behavior near the origin is fitted well by a Gaussian model (blue dashed lines). **Therefore, calculating the experimental semivariogram based on slice-by-slice porosity is an appropriate tool for variogram interpretation, a tool that for the best of our knowledge was not used in previous studies using CT data**.

- Furthermore, it was stated by the reviewer that "The computation of the experimental variograms in terms of retained "bins" and "lags" must be probably adjusted". We followed the reviewer's suggestion to plot semivariograms with a lag distance of larger support volumes (Figure 12**aa** below). When using lag separations at the scale size of the fluctuations (125 µm, in black), the fluctuations are only dampened and smoothed. **We think, therefore, that regularizing is not needed for semivariogram calculation based on slice-by-slice porosity of CT data**.

- Concerning the "High-frequency fluctuations": Due to the absence of a nugget effect, the high-frequency fluctuations are assumed to be related to a true geological-derived spatial variability in the sample, rather than to a non-regularized image (see above). The average distance from peak-to-trough in the semivariogram is about 0.1 mm, with a similar value to the range of correlation that was calibrated with the Gaussian model for all samples and in all directions. From petrographic imaging of the sandstone sample, we relate these fluctuations in the semivariogram to changes in porosity influenced by the grain packing / sorting. Within the semivariograms, an additional structure with high variability can be observed (e.g. 12_new_c, 18_new_a-c, below), causing the scale of fluctuations in the semivariograms (produced by the grain packing) to be "overshadowed". The structure that overshadows has a range of correlation on the mm-scale and relates to the more "macroscopic" inhomogeneity in each direction. The analysis and our interpretation for each sample are the following:

  ➢ For S1, Figure *12_new_c* shows mm structure only along z-direction. In this sample that structure relates to the mm-scale layering that was observed from thin section microscopy. The mm-structure that is derived in z-direction and its absence in x- and y- directions implies a distinct anisotropy.

  ➢ For S2 Figure *18_new_c* mm-scale structures are observed in x-, y- and z- directions. However, in the z-direction the calibrated range of correlation of the hole-effect model is 2 mm, whereas in x- and y- directions the range is 1 mm. This sample contains a clayey matrix, which is distributed in the sample in a patchy behavior. Accordingly, the structures observed and calculated from the semivariogram are related to this geological phenomenon. The larger heterogeneity in z-direction again implies a distinct anisotropy. The longer range in z-direction is related to layering, as derived from the high (negative-) correlation between porosity and clay matrix in z-direction and no correlation in the x- and y- directions (Figures 15-16 in the manuscript).

  ➢ For S3 *Figure 14*_new, the semivariograms do not show a mm-scale structure in any direction. Results are thus solely related to the grain packing. It implies that no other structure exists in the domain under investigation, and that this sample is relatively homogenous compared to the others.

[Figure]

*Figure 12_new: Semivariograms of S1 in a) x-direction, b) y-direction and c) z-direction. z-score transform was applied to the histograms (Figure 11d). For z-direction it was applied before (in black) and after (in red) the trend removal (Figure 11c). The variogram analytical models (Gaussian and hole-effect) are also displayed (dashed blue line), modelled with nugget set to zero. d-f) zoom-in on short lag distances to assess the behavior near the origin and the goodness of the fit.*

[Figure]

*Figure 12aa: Experimental semivariograms calculated with 3 different bin sizes: 5, 50 and 125 μm, in a) X-, b) Y- and c) Z- axis, for sample S1.*

[Figure]

*Figure 14_new: Semivariograms of S3 in a) x-direction, b) y-direction and c) z-direction. The experimental semivariogram was modelled using Gaussian model (dashed blue line). The semivariogram models are also displayed, modelled with nugget set to zero d-f) zoom-in on short lag distances to assess the behavior near the origin and the goodness of the fit.*

[Figure]

*Figure 18_new: Semivariograms of S2 in a) x-direction, b) y-direction and c) a-direction. z-score transformation was applied to the histogram, before (black) and after (red) the trend removal. The experimental semivariogram was also modelled using nested Gaussian, spherical and hole-effect models*

*(dashed blue line) with nugget set to zero. d-f) zoom-in on short lag distances to assess the behavior near the origin and the goodness of the fit.*

Reviewer's Comment 2:

For the same graphs, it would help to have the experimental variograms plotted as points at the centres of each actually computed bin instead of (what it seems) regularly spaced points, which I can't tell if they correspond to the actual computed bins or if it's a graphical artifact.

Authors' Response 2:

Actually, the calculated values are now displayed by points (in red, Figure 12_new, 14_new, 18_new). The reason it seems like a line is that the lag separations (or bin size) is the size of a voxel 5 µm. Within the range of between the minimal and maximal lags (5 µm and 3000) there are 600 points, thus look continuous and seemed like a line demonstrated previously. The points can be seen when using the zoom-in tool. Thus, one can see that semivariogram values refer to a lag distance which excludes any graphical artifacts.

Reviewer's Comment 3

The z-direction in fig. 12(c) shows a clear nugget effect, I don't see why the authors don't acknowledge that. The same applies to fig. 18(b).

Authors' Response 3:

Please see Figure *12_new_f,* which zooms-in to assess the behavior near the origin and shows that there is no nugget effect, but a parabolic behavior (S-shaped) near the origin.

Reviewer's Comment 4:

The de-trending of the sample S1 if I understand correctly, has been done just subtracting the average of the porosity, and not using the "external drift" represented by the cement vol. percent, which is actually provided by figure 20 and is an excellent observation. So, when doing a linear detrend the authors should use a linear model based on the cement volume-% in direction z. Even better however, since it's clear from figure 11(c) that we are in presence of a clear stratification with a period of little less than 4 mm in z-direction (corresponding indeed to the minimum value of the variogram for distances at little less than 4 mm), it's clear from the beginning that a linear de-trend would not be sufficient in this case. A solution would be to apply here a higher order de-trend (polynomial but also smoothing splines or loess...), so to focus on the fluctuations around the middle and eliminate the hole effect all together.

Authors' Response 4:

- The de-trending in z-axis was performed by subtracting the "external drift" as shown in the histogram 11d (in the former version of the manuscript). Figure 11C (red, right y-axis) shows the de-trended residual porosity, slice-by-slice profile. The de-trending was performed based on the trend observed in slice-by-slice porosity (black, left y-axis). The de-trend functions chosen were either 1st or 2nd order polynomial fit, when the 2nd order one was chosen if the improvement compared to the 1st order fit was significantly better.

- The reviewer suggests using the cement percentage in S1 for de-trending the slice-by-slice porosity, by calculating the cement trend. The cement in S1 is observed by microscopy as meniscus bridging and grain coating made of flakes at 1-10 µm scale. This size is in the order of the imaging resolution (scans of 2.5 and 5 µm voxels grid), not high enough for quantification of the exact characteristics (volume, type, grain sizes), therefore relying on cement content as a property which is correlated with porosity encompasses much uncertainty and therefore should not be considered. Sample S2 with a prevalence of clay matrix (Figure 15 in the manuscript), however, does show a negative correlation in z-direction (Figure 16c), which relates to the deposition processes, but no correlation in x- and y- directions. Therefore, we think that doing the de-trending for porosity based on the clay or cement is not the way, but using the porosity for de-trending, as we previously used.

Reviewer's Comment 5:

Concerning figure 19 and the whole study about it, it seems to me that it just refers to subsets of the original sample, in which the variograms were still computed slice by slice, whereas I initially suggested to use 3D "supports" for the computation of the experimental variograms, and not restrictions of the domain. I am not convinced that this part adds valuable informations to the analysis at the moment.

Authors' Response 5:

As we understand, the reviewer suggests performing a 3D variography based on single pixels and not slice by slice porosity. We tested the 3D variography for the CT data performing the additional coding, for a histogram used for computation is binary – voxel values are either '1' (pore) or '0' (grain/matrix). We are interested in calculating the correlation length of the pores, therefore only pore voxels are considered when voxels were chosen randomly. Because the entire dataset computationally is too heavy (more than 1000x1000x1000 voxels, spatially continuous data), we used in the experimental semivariogram computation only a part of the total possible pairs (we used $10^6$ pairs in each direction). The semivariogram results for S1 are shown in Figure 21 below, zooming in at different lag distances.

➢ For short distances (Figure 21a) for all directions the range of correlation is approximated to about 0.1 mm, similar to range calibrated from the semivariogram computed from slice-by-slice porosity (Figure 12_new above). This structure is related to the grain packing.
➢ For intermediate maximal lag distances (Figures 21 b,c), no structure is observed, in contrast to the layering structure of mm-scale that was observed from the slice-by-slice porosity in z-direction (Figure 12c). Figure 21 is presented just as a clarification for the reviewer.

- Figure 19 in the manuscript is removed (along with other figures related to the variography).

[Figure]

[Figure]

[Figure]

a          b          c

*Figure 21: Semivariogram for S1 measured on voxel-based binarized data. Randomly chosen voxels are pores. For interpretation, plots are zoomed-in on the ranges: a) 0-250 µm, b) 0-2000 µm and c) 0-6300 µm. Sill=1.12. Number of voxel pairs used is ~$10^6$.*

Reviewer's Comment 6:

Given the flaws of this variographic analysis it's not surprising to me that the "apparent", interpreted range is ten times smaller than the classical REV determination. The authors should consider either doing it properly, possibly involving some experienced geostatistician in the analysis, or else leave it all together.

Authors' Response 6:

We agree with the reviewer, and removed all of the mentioned variography. Nevertheless, the semivariogram is a useful tool to estimate distances of correlation and lack of correlation between measurements. Determining this range of correlation is useful to estimate the scale of *heterogeneity*, and is not directly related to the *directional REV*. It is more an apparent range, influenced by cementation, clay minerals and grains. For S3, this apparent range is shorter compared to the REV determined by the classical approach. The shorter apparent range from the semivariogram is in our opinion clearly related to the grain packing, the only structural phenomenon at the size of the sample domain under investigation. As a rule of thumb, one would use at least four (4) grain diameters as REV edge size.

Reviewer's Comment 7

I salute however the new simulations of permeability within the whole sample S3. In my opinion this part should not be in appendix but discussed in more detail in the main text.

Authors' Response 7:

The simulations in sub-volumes and a full volume of S3 appear now at the main text (Figures 15, 16) and discussed in lines 858-862, 990-999, 1109-1155 in our revised manuscript.